



# 1  Exploring the sensitivity of the Northern Hemisphere ice sheets at the
# 2  last two glacial maxima to coupled climate-ice sheet model parameters

Violet L. Patterson[1], Lauren J. Gregoire[1], Ruza F. Ivanovic[1], Niall Gandy[2], Stephen Cornford[3], Jonathan
Owen[4], Sam Sherriff-Tadano[5], Robin S. Smith[6]
[1]School of Earth and Environment, University of Leeds, Leeds, UK
[2]Department of the Natural and Built Environment, Sheffield Hallam University, Sheffield, UK
[3] School of Geographical Sciences, University of Bristol, Bristol, UK
[4] School of Mathematical and Physical Sciences, University of Sheffield, Sheffield, UK
[5]Faculty of Science, University of the Ryukyus, Okinawa, Japan
[6] NCAS, Department of Meteorology, University of Reading, Reading, UK
*Correspondence to*: Violet L. Patterson (ee17vp@leeds.ac.uk)
**Abstract.** Simulations of past periods are useful for testing the ability of numerical models to simulate ice sheet changes under
significantly different climate conditions to present day. This can help improve projections of future sea level rise made by
these same models and avoid over-tuning to particular (e.g. modern) stationary climate conditions. The Last Glacial Maximum
(LGM; ~21 thousand years ago (ka)) has been extensively used for this purpose since it is relatively well constrained by
empirical evidence. However, less is known about the Penultimate Glacial Maximum (PGM; ~140 ka) and why the vast ice
sheets covering much of the Northern Hemisphere (NH), differed to the LGM. The answer likely lies, at least in part, in the
different orbital configurations between the two periods, and the resulting impact on climate-ice sheet interactions.
Here, we perform and compare the first large ensembles of coupled climate-ice sheet (FAMOUS-BISICLES) simulations of
the LGM and PGM to better understand how NH ice sheets interact with the climate and quantify how sensitive the simulations
are to the choice of uncertain model inputs, including physical parameter values. Specifically, we vary 12 uncertain parameters
that control the model representations of ice sheet albedo, ice dynamics and climate. The ensembles are evaluated against
palaeo-evidence of global mean temperature, ice volume and extent to calibrate the model and find combinations of parameters
that simultaneously yield plausible ice sheets and climates for both periods. The sensitivity of the North American ice sheet
and the Eurasian ice sheet during the LGM and PGM, to each of the 12 parameter values, is explored using Gaussian Process
emulators to perform a Sobol sensitivity analysis. From the whole ensemble, we find two simulations that meet our evaluation
constraints for the LGM ice sheets. The parameter values that influence the albedo of the ice sheet have the largest influence
on the resulting ice sheet volumes, but several other parameters display different sensitivity indices depending on the ice sheet
(North American versus Eurasian) and time period (PGM versus LGM). This includes parameters that affect the cloud liquid
water, lapse rate, basal sliding and downscaling elevation heights.



## 1 Introduction

During glacial periods of the last 800,000 years, large ice sheets built up over the Northern Hemisphere (NH) continents (Ehlers et al., 2018) impacting the climate through their interactions with atmospheric circulation, oceanic circulation and the energy budget (Lambeck et al., 2014; Scherrenberg et al., 2023b). However, the evolution of the NH ice sheets differed between each glacial period leading to different geometries at the glacial maxima, the periods during the glacials in which global ice volume is at its largest and global mean sea level is at its lowest (Ehlers et al., 2018).

Geological evidence and numerical simulations of the last two glacial maxima, the Penultimate Glacial Maximum (PGM; ~140 ka) and the Last Glacial Maximum (LGM; ~21 ka), for example, suggest very different configurations of the North American ice sheet (NAIS) and the Eurasian ice sheet (EIS) (Svendsen et al., 2004; Colleoni et al., 2016; Batchelor et al., 2019) despite similarities in Greenhouse Gas (GHG) concentrations ($CO_2$ ~ 190 ppm), global average insolation and global ice volume (~ 130 meters sea level equivalent (m s.l.e.)) (Berger and Loutre, 1991; Loulergue et al., 2008; Rabineau et al., 2006; Masson-Delmotte et al., 2010; Bereiter et al., 2015; Rohling et al., 2017). Geomorphological evidence suggests that the extent of the Penultimate EIS could have been ~50% larger than during the Last Glacial Cycle and expanded 200 km further south and 1000 km further east in Siberia (Batchelor et al., 2019; Knies et al., 2001; Svendsen et al., 2004). However, there are large uncertainties in its maximum extent at the PGM since there is evidence of two major ice advances in Europe, the more extensive Drenthe (~160 ka), which was followed by partial melting and sea level rise ~157-154 ka under increasing summer insolation, and then a readvance after 150 ka during the less extensive Warthe (Hughes and Gibbard, 2018). Thus, current reconstructions of the maximum may incorrectly incorporate previous advances during MIS 6 (195-123 ka) (Ehlers et al., 2018; Margari et al., 2014; Svendsen et al., 2004).

Since the volume of ice sheets cannot be directly inferred from empirical evidence, it must be indirectly estimated from datasets such as relative sea level proxies through glacial isostatic adjustment (GIA) inversion modelling and numerical ice sheet modelling (e.g. Lambeck et al., 2006; Tarasov et al., 2012; Rohling et al., 2017). Consequently, there is even larger uncertainty in volume estimates than there are in extent estimates. Nonetheless, ice volume estimates support the ice extent-derived evidence that EIS volume was indeed larger at the PGM, with most estimates ranging from ~40-70 m s.l.e. compared to ~13-24 m s.l.e. at the LGM (Lambeck et al., 2006; Peyaud, 2006; Pollard et al., 2023; Rohling et al., 2017; Simms et al., 2019; Tarasov et al., 2012).

In contrast, whilst there is some evidence that, during the PGM, the NAIS extended slightly further south in the regions known today as Illinois and Wisconsin (Batchelor et al., 2019; Hughes and Gibbard, 2018), most available evidence suggests that the NAIS was smaller in extent and volume compared to the LGM. This includes relative sea level assessment studies (e.g. Rohling et al., 2017), reduced ice rafted debris layers in the North Atlantic (pointing to reduced iceberg discharge from the Hudson Bay region; Hemming, 2004; Naafs et al., 2013; Obrochta et al., 2014), climate and ice sheet modelling studies (Abe-Ouchi et al., 2013; Colleoni et al., 2016; Wekerle et al., 2016) and GIA modelling studies (Dyer et al., 2021; Wainer et al., 2017). The relative lack of geomorphological evidence of the PGM NAIS further supports the hypothesis that PGM NAIS was smaller





than LGM NAIS because it implies a larger ice advance at the LGM destroyed most traces of the previous glacial maximum
(Dalton et al., 2022; Dyke et al., 2002; Rohling et al., 2017). Therefore, the footprint of the PGM NAIS remains very uncertain,
while LGM NAIS ice extent is relatively well constrained from a range of glacial geological evidence, which has been updated
in recent years (e.g. Dalton et al., 2020). As with the EIS, the volume of the NAIS is more difficult to assess from empirical
evidence and mostly relies on modelling, which estimates it at being between ~39-59 m s.l.e. at the PGM compared to ~68-88
m s.l.e. at the LGM (Rohling et al., 2017; Simms et al., 2019).
The differences in the shape and size of the ice sheets between the LGM and PGM are not well understood. They result from
complex interactions occurring between different components of the earth system (e.g. atmosphere, ocean, ice sheets, and solid
earth) leading up to and at the glacial maximum. Despite similar levels of average global incoming solar radiation between the
LGM and PGM, the seasonal and latitudinal patterns differed between the two periods, as did its evolution prior to the maxima,
as a result of different orbital situations (Berger, 1978; Berger and Loutre, 1991). The orbital forcing, along with concentrations
of GHGs, would have altered the radiative balance between the periods. As well as affecting the ice sheet evolutions directly,
this also would have influenced the sources and pathways of moisture advection (Hughes and Gibbard, 2018; Krinner et al.,
2011; Rohling et al., 2017), sea surface temperatures (SSTs) and sea ice concentration (Clark et al., 2009; Colleoni et al., 2011;
Kageyama et al., 1999; Kageyama and Valdes, 2000), vegetation distribution (Colleoni et al., 2009b; Kageyama et al., 2004;
Stone and Lunt, 2013), dust deposition (Colleoni et al., 2009a; Krinner et al., 2006; Naafs et al., 2012) and pro-glacial lake
coverage (Colleoni et al., 2009a; Krinner et al., 2004), which all have important feedbacks onto the climate.  Additionally,
feedbacks on the climate from the ice sheets themselves are very important in regulating ice sheet surface mass balance (SMB),
for example through the influence of the ice-albedo and temperature-elevation feedbacks on surface temperature and energy
balance (Abe-Ouchi et al., 2007; Patterson et al., 2024), and interactions between atmospheric and oceanic circulation, surface
temperature and precipitation patterns (Beghin et al., 2014, 2015; Liakka et al., 2012). Some studies have also concluded that
the topography of the NAIS had a large influence in the size and configuration of the EIS through its effect on the jet stream
and stationary waves (Beghin et al., 2015; Liakka et al., 2016).
Direct observations of processes occurring during glacial cycles are not available and while proxy evidence can provide
important constraints on how the ice sheets changed, it cannot reveal the mechanisms behind these changes. Numerical
modelling is therefore required to understand the response of the NH ice sheets to external and internal forcings and unpack
why they differed between glacial periods. This is an important source of information in the context of understanding how ice
sheets may respond to future climate change (Gregory et al., 2012). Currently there are large uncertainties in projections of
future sea level rise (Edwards et al., 2021; Intergovernmental Panel On Climate Change, 2021) mainly as a result of limited
knowledge of several important ice sheet processes, such as non-linear behaviours of the ice sheet system, and climate and ice
sheet interactions (Golledge et al., 2019; Gregoire et al., 2012; Kopp et al., 2017). Simulations of past periods can help improve
our understanding of these processes as well as help evaluate and refine the numerical models used for these projections
(Braconnot et al., 2012; Gandy et al., 2018; Harrison et al., 2016; Masson-Delmotte et al., 2013; Schmidt et al., 2014). The
LGM has been extensively used for this purpose because the climate and ice sheet states are relatively well constrained by





empirical evidence and thus allow evaluation of model performance, helping constrain climate and ice sheet models and future
sea level projections (Gandy et al., 2023; Ziemen et al., 2014). Furthermore, the EIS has large marine based sectors in the
Barents-Kara and North Sea regions and thus it is often considered an analogue of the current West Antarctic Ice Sheet.
Modelling and identifying the mechanisms responsible for the different EIS evolutions might help with understanding the
processes in effect in West Antarctica and its vulnerabilities to climate change (van Aalderen et al., 2023; Gandy et al., 2018).
Many previous studies simulating the NH LGM and PGM climate and ice sheets have treated the components independently.
Either prescribing the ice sheets as a boundary condition in a climate model, which neglects any affects the climate has on the
ice sheets (Beghin et al., 2015; Colleoni et al., 2016; Hofer et al., 2012; Merz et al., 2015; Ullman et al., 2014), or forcing ice
sheet models with climate output from GCMs, which introduces large uncertainties depending on the model used and can
produce unrealistic ice sheets (Abe-Ouchi et al., 2013; Alder and Hostetler, 2019; Charbit et al., 2007; Gregoire et al., 2016;
Niu et al., 2019; Scherrenberg et al., 2023b; Wekerle et al., 2016; Zweck and Huybrechts, 2005). Thus, the use of directly
coupled climate-ice sheet models to perform these simulations will explicitly resolve some of these important feedbacks and
interactions between the climate and the ice sheets, reducing some of the uncertainties and inconsistencies caused by
prescribing one of the components, and provide a better understanding of these processes (Abe-Ouchi et al., 2013; Niu et al.,
2019; Quiquet et al., 2021; Ziemen et al., 2014).
Recent developments have allowed the two-way coupling between GCMs and ice sheet models, but previous studies using
this method have focused on just one time period and/or one ice sheet and there have so far been no coupled GCM-ISM
simulations of the NH ice sheets at the PGM (Gandy et al., 2023; Gregory et al., 2012; Patterson et al., 2024; Quiquet et al.,
2021; Sherriff-Tadano et al., 2024; Ziemen et al., 2014). Additionally, it has been shown that uncertainties in certain model
parameters can have a large influence on the resulting ice volumes simulated by the coupled model through altering the strength
of important climate-ice sheet feedbacks (Gandy et al., 2023; Patterson et al., 2024; Sherriff-Tadano et al., 2024). Patterson et
al., (2024) evaluated a range of model parameter values based on whether they produced plausible NAIS configurations for
both the LGM and PGM. However, the different processes operating on the Eurasian ice sheet (see Sect. 2.1), the interactions
that may occur between both ice sheets and the use of a different ice sheet model with more advanced physics and an updated
experimental design, require additional uncertainty quantification to be carried out through a large ensemble analysis, to re-
evaluate the collection of parameter combinations that yield model output consistent with observation data (up to the assessed
uncertainties), referred to as the 'Not Ruled Out Yet' (NROY) parameter space (Williamson et al., 2013).
The aim of this work is therefore to perform and compare ensemble simulations of the NH ice sheets at the LGM and PGM
using a coupled climate-ice sheet model (FAMOUS-BISICLES). After performing some sensitivity tests to optimise the model
for ice streaming in the NH ice sheets, we assess the ability of the model to produce reasonable simulations of both the NAIS
and EIS for both periods. We evaluate the impact of uncertainty in model parameters on the resulting ice sheets and whether
both ice sheets show similar sensitivities to the parameters. The model is evaluated against an implausibility metric based on
ice sheet volume and extent data, and the representation of ice streams is assessed.





## 2 Methods

### 2.1 Models

The climate model used in this study, FAMOUS, is sufficiently efficient that it is suitable for running long (multi-millennial) palaeo simulations (e.g. Gregory et al., 2012; Gregoire et al., 2012; Roberts et al., 2014; Dentith et al., 2020) and large ensembles for uncertainty quantification (Gandy et al., 2023; Gregoire et al., 2011; Sherriff-Tadano et al., 2024), whilst still resolving the same complex processes as represented in an Atmosphere-Ocean General Circulation Model (AOGCM). It is based on HadCM3 AOGCM (Gordon et al., 2000; Pope et al., 2000) but has half the spatial resolution and a longer time-step, thus requiring only 10 % of the computational costs of the parent GCM.

We use the atmospheric component of FAMOUS, which is a hydrostatic, primitive equation grid point model with a horizontal resolution of 7.5° longitude by 5° latitude with 11 vertical levels and a 1-hour time step (Williams et al., 2013). FAMOUS can also be run coupled with a dynamical ocean (e.g. Dentith et al., 2020), however, in this study, we prescribe sea surface temperatures and sea ice (see Sect. 2.3.1). The land surface scheme MOSES2.2 (Essery et al., 2003) is used to represent land processes on a set of sub-grid scale tiles in each grid box representing fractions of nine different surface types, including land ice (Smith et al., 2021).

This study uses a version of FAMOUS developed to have bi-directional coupling to an ice sheet model (FAMOUS-ice; Smith et al., 2021) accounting for the mismatch between the atmosphere and ice sheet grid sizes by using sub-grid scale elevation tiles. The atmospheric surface air temperature and long wave radiation is calculated in FAMOUS at the mean elevation within each grid cell and for ice sheet grid cells, these quantities are downscaled onto 10 vertical "ice tiles" with different elevations; 100 m, 300 m, 550 m, 850 m, 1150 m, 1450 m, 1800 m, 2250 m, 2750 m, 3600 m. The air temperature downscaling is done by using a constant lapse rate (*tgrad*) to adjust for the differences in the elevation between each tile and the mean elevation, and humidity and downwelling longwave are adjusted to be consistent with the temperature adjustment. No downscaling is applied to precipitation and shortwave radiation in this version of the model. The surface energy fluxes and SMB are calculated on the 10 ice tiles based on the energy budget equation and a multi-layer deep snowpack model. Then the SMB is passed onto the ice sheet model, which projects and linearly interpolates this coarse 3D lat-lon SMB field onto the higher resolution ice sheet surface. The resulting changes in ice extent and surface elevation simulated by the ice sheet model are passed back to FAMOUS to update the fraction of ice present within each ice tile and the orography fields. Within FAMOUS, the mean of the surface fluxes weighted by ice fraction within the ice tiles sets the land-atmosphere exchanges within FAMOUS. In this study, this process is run at 10 times ice sheet model acceleration meaning one year of climate integrated in FAMOUS is used to force 10 years of ice sheet integration in the dynamical ice sheet model before the ice cover and orography fields are passed back (Gregory et al., 2020).

In a previous study, Patterson et al., (2024) used FAMOUS coupled to the Glimmer ice sheet model to simulate the North American ice sheet. However, the coarse resolution and the use of Shallow Ice Approximation (SIA) in the Glimmer ice sheet model used in that study does not resolve the small-scale processes or longitudinal stresses required to accurately simulate ice





stream evolution or grounding line migration. Whilst these processes are not as important to capture in an equilibrium spin up
of a continental size terrestrial ice sheet, such as NAIS, they have a large influence on the behaviour, configuration and stability
of a marine ice sheet (Hubbard et al., 2009; Pattyn et al., 2012; Stokes and Clark, 2001). In particular, the Eurasian ice sheet
has many ice streams within marine sectors (e.g. North Sea and Barents Sea) that are vulnerable to processes that may cause
instabilities of retreat, for example Marine Ice Sheet Instability (MISI), and are likely to have been important in its evolution
and deglaciation (Kopp et al., 2017). These processes are similar to those in operation today in West Antarctica, currently
forming a large source of uncertainty in future sea level projections (van Aalderen et al., 2023; Alvarez-Solas et al., 2019;
Edwards et al., 2019; Gandy et al., 2019, 2021; Petrini et al., 2020).
BISICLES is well suited to simulating marine ice sheet dynamics due to its use of the L1L2 physics for approximating the
sliding and flow of the ice sheet, instead of SIA (Cornford et al., 2013). The L1L2 approximation is a variant of Glen's flow
law that includes longitudinal and lateral stresses and approximates vertical shear strains in vertically integrated models
(Schoof and Hindmarsh, 2010). This makes it able to represent ice-shelves and fast-flowing ice streams (Hindmarsh, 2009).
Additionally, some ice sheet processes, such as ice streaming and grounding line migration, require high resolution to simulate
accurately. BISICLES enables this to be feasible in millennial scale and large ensemble simulations through its adaptive mesh
refinement (AMR). Where required, the model can simulate at high resolution, whilst the rest of the domain (i.e. the slower
moving interior of ice sheets) remains at a lower resolution, thus increasing the efficiency of the model (Cornford et al., 2013).
With these features, BISICLES is a model well suited to simulate the past evolution of marine ice sheets such as the Eurasian
ice sheet. It also allows for better physical accuracy in the representation of ice streams within the North American ice sheet.
BISICLES has previously been used to successfully simulate the ice streams and retreat of the marine based British-Irish Ice
Sheet at the Last Deglaciation (Gandy et al., 2018, 2019, 2021), the final retreat of the NAIS during the early Holocene (Matero
et al., 2020), produce an initial condition of the present-day Greenland Ice Sheet (Lee et al., 2015) and model the future
evolution of the Antarctic Ice Sheet (Cornford et al., 2015; Siahaan et al., 2022). Additionally, FAMOUS-BISICLES has been
used to explore the sensitivity of the NAIS and Greenland Ice Sheet at the LGM to model parameter values through large
ensemble analysis (Sherriff-Tadano et al., 2024).
We use the updated version of BISICLES developed by Gandy et al., (2019), which implements a pressure limited basal sliding
law that is sensitive to the presence of till water. This is mostly found to be applicable near the grounding line and the inclusion
of the Coulomb sliding law has been shown to have an effect on ice sheet stability in models, with greater grounding line
retreat occurring in simulations that include this law than those without (Nias et al., 2018; Schoof, 2006; Tsai et al., 2015).The
upper surface temperature boundary condition in the ice sheet model (surface heat flux) is determined by the climate model
and the basal boundary condition (basal heat flux) is set as a constant flux ($3 \times 10^6$ J a$^{-1}$ m$^{-2}$). The effective pressure, and
therefore the basal sliding, depends on the basal water pressure and thus the depth of the till water layer. Once the englacial
drainage water fraction *(w)* grows beyond a certain value (0.01) it is drained to a till layer at a rate proportional to the water
fraction, up until a maximum water fraction (0.05). The till water is then transported elsewhere by the basal hydrology model
(Van Pelt and Oerlemans, 2012). It is lost vertically at a rate proportional to the till water depth which is determined by the



specified till water drain factor (*drain*). A maximum till water thickness of 2 m is set following previous studies (Bueler and
van Pelt, 2015; Gandy et al., 2019; Moreno-Parada et al., 2023). A recent comparison study by Drew and Tarasov (2023)
shows that this simplified 'leaky bucket' hydrology scheme produces similar results to more complete models over centennial
or longer timescales and continental scale ice sheets. Additionally, the implementation of this basal sliding scheme coupled
with this hydrology parameterisation allows the simulation of spontaneous ice stream generation and evolution (Gandy et al.,

203  2019, 2021).

The upper surface thickness flux (i.e. accumulation/melt) is calculated by the climate model and the lower surface (basal)
thickness flux (i.e. oceanic melt) is set to zero for grounded ice and is proportional to the SSTs for floating ice, according to
the linear relationship;
$$Subshelf\ melt\ rate\ (myr^{-1})\ =\ c(T_{ocn} - T_f) \tag{1}$$
Where c is a constant, $T_{ocn}$ is the prescribed sea surface temperature and $T_f$ is the freezing point of seawater, assumed to be -
1.8 °C at the surface (Alvarez-Solas et al., 2019; Beckmann and Goosse, 2003; Gandy et al., 2018; Martin et al., 2011; Rignot
and Jacobs, 2002). Since the freezing point of sea water varies with depth of the ice shelf base and with salinity, and the surface
temperatures are used rather than subsurface, this is a highly idealised parameterisation. In addition, many studies have found
a quadratic relationship to be a better fit to present-day observations (e.g. DeConto and Pollard, 2016; Favier et al., 2019;
Holland et al., 2008). However, the lack of constraints on ice shelves, ocean temperatures, and sub-shelf melt rates for the
periods covered in this study makes this a large source of uncertainty in our modelling. In this context, it is preferable to choose
a simple linear representation of sub-shelf melt over a more complex quadratic relationship. We account for this uncertainty
in the wide range of sub-shelf melt constant (*c*) values used (1 – 50 m yr$^{-1}$ °C$^{-1}$). This relationship produces an average sub-
shelf melt rate across the ice shelves of between around 1.6 – 28 m yr$^{-1}$, which are not unrealistic when compared to the
estimates from present-day Antarctica of 0 – 43 m yr$^{-1}$ (Depoorter et al., 2013; Jourdain et al., 2022; Rignot et al., 2013).
However, some regions in some simulations display very large rates of 100s of metres per year.
Glacial isostatic adjustment (GIA) of bedrock topography due to changes in the ice sheet load is included through coupling
BISICLES to a simple Elastic Lithosphere Relaxing Asthenosphere (ELRA) model, which approximates this response by
assuming a fully elastic lithosphere above a uniformly viscous asthenosphere (Kachuck et al., 2020). A relaxation time of 3000
years is applied in this model based on previous studies (Pollard and DeConto, 2012). This method does not account for
changes in the gravitational pull that ice sheets exert on sea level or adjustments in Eustatic sea level caused by changing
global ice sheet volume (e.g. Gomez et al., 2010).
Sherriff-Tadano et al. (2024) found that some of the FAMOUS-BISICLES simulations of the NAIS at the LGM exhibit a
strong local melting of the ice sheet from parts of the interior. This phenomenon is caused by warm temperature biases over
the ice sheet interior in the atmospheric model, which are amplified by the downscaling method and a positive height-mass
balance feedback. A similar temperature bias was pointed out by Smith et al., (2021) using the same model under the modern
Greenland ice sheet, which produced a higher Equilibrium Line Altitude (ELA) (around 2 km high in places) compared to a
high-resolution regional atmospheric model (at about 1 km high). The warm temperature bias comes from the low resolution





of the atmospheric model. In reality, a very cold atmospheric layer often forms at the surface of the ice sheet, especially in the
interior, which induces a stable boundary layer and isolates the cold surface from the ambient warm air. However, a global
climate model cannot resolve the effect of the stable boundary layer and overestimates the exchange of heat between the
surrounding atmosphere and the ice sheet surface. As a result, FAMOUS overestimates the temperature in the ice sheet interior
and causes a high ELA bias, which results in surface melt.
Here, we take a practical approach to mitigate the effect of the warm temperature bias in FAMOUS. This is done by modifying
the height adjustment of atmospheric surface temperature to the ice tiles through the introduction of a new parameter in the
model, *elevcon,* which is intended to make the parts of the ice sheet surface well inside the margins colder. Appendix A
includes a description of how the *elevcon* parameter is implemented and works to affect the surface temperature and SMB
during height correction, and of sensitivity experiments performed to validate the effect of different values of *elevcon* on the
modern and LGM ice sheets and climates. Since the optimal value of this adjustment is uncertain, we include *elevcon* in the
ensemble as a varied parameter value, between the range of 1 and 1.5 (0-50 %). These values were chosen based on testing
that showed that a value of 1.5 produced an equilibrium line altitude height that represents an upper limit determined by
empirical data (Fig. A1).
**2.2 Ice dynamics in BISICLES**
It has been established that ice streams exert an important control on the behaviour and geometry of an ice sheet and therefore
it is crucial that in our study, the simulated location and dynamics of at least the major ice stream features, are consistent with
reconstructions. Gandy et al. (2019) highlighted that the most important model ingredient necessary to successfully model ice
streams is the representation of idealised subglacial hydrology. The till water layer coupled with the Coulomb sliding law
described in Sect. 2.1 is crucial for the spontaneous generation of ice streams. However, this scheme is highly sensitive to the
drainage and temperature structure of the ice sheets. Inadequate consideration of these factors can lead to a poor representation
of ice streams (e.g. Sherriff-Tadano et al., 2024). Therefore, we perform a spin up of BISICLES that results in the internal
temperatures of the ice sheet being more conducive for ice stream generation over shorter integration times. We also perform
sensitivity tests varying the level of refinement of the ice streams and the rate of till water drainage to find an optimum set-up
that balances computational cost with the representation of ice dynamics. These methods are described in the following
sections.
**2.2.1 Temperature spin-up**
The internal temperature of ice sheets is an important factor in controlling the deformation, rheology and velocity of the ice
due to the temperature dependence of the sliding law and enthalpy scheme (Blatter et al., 2010). The ice sheets start with a
uniform internal temperature of 268 K and it can take tens of thousands of years for the process of cold ice advection from the
interior and heat conduction from the bed to occur and reach an equilibrium, which is important for the formation of ice streams
(Fyke et al., 2014; Heine and Mctigue, 1996). Thus, we perform ice sheet model only spin-ups for the LGM and the PGM to



allow the ice sheet internal temperatures to reach close to equilibrium. This temperature profile is then used as the internal ice
sheet temperature in the initial condition for the sensitivity tests (Sect. 2.2.2 and 2.2.3) and coupled simulations.
The spin ups were run at 32 km resolution for 20,000 years using single surface mass balance and surface temperature fields
taken from a FAMOUS-BISICLES equilibrium simulation that used climate model parameters identified to be NROY in
simulations of the NAIS by Patterson et al., (2024), default ice sheet model parameters and an *elevcon* value of *1.2* (Fig. B1).
The initial ice sheet configurations were the same as used in the coupled simulations (described in Sect. 2.3.1; Fig. 1). The
sliding law was set to a temperature dependent Weertman sliding without till water dependent Coulomb sliding enabled since
the bulk of the temperature field is not affected much by Coulomb sliding near the coast. The resulting temperature profiles
are shown in Appendix B (Figs. B2 and B3).

**2.2.2 Drain factor sensitivity tests**

In their study, Sherriff-Tadano et al., (2024) used much higher values of *drain* (0.2-0.6 m yr$^{-1}$) than has typically been used in
previous studies (0.001-0.005 m yr$^{-1}$; Gandy et al., 2019; Kazmierczak et al., 2022; Moreno-Parada et al., 2023). This was to
prevent large till water depths leading to too large velocities across the entire ice sheet and long simulation times, as high
velocities require more iterations and smaller timesteps to solve. This resulted in the till water drainage outpacing the supply
and thus very small till water depths, leading to mostly Weertman sliding across the whole ice sheet.
Slow till drainage (low values of *drain*) can lead to isolated regions of fast flow, > 50 km yr$^{-1}$, which have a disproportionate
effect on simulation time. To prevent this we introduce an artificial drag term rising with the fourth power of ice speed and
calibrated to be negligible for ice speeds below 1 km yr$^{-1}$. This drag factor is also used in the coupled simulations throughout
the rest of this study. We then perform sensitivity tests with different values of *drain* spanning the range 0.001-0.06 m yr$^{-1}$ but
all other factors kept constant. The results of some of these tests are shown in Fig. C1. Values of *drain* above 0.05 prevent
much of the coulomb sliding at the coasts and the representation of some of the major ice streams, particularly the Hudson
Strait Ice Stream, is poor. Low values usually used in ice sheet models (0.001-0.005) cause too large velocities and ice streams
that remove much of the ice sheet, especially in Eurasia. Therefore, in this study, we implement a range of 0.01-0.05 to cover
values just below the default till water supply rate of 0.02, to where no coulomb sliding occurs. For studies that seek to examine
ice streaming of the glacial maximum ice sheets, we would recommend performing additional sensitivity tests that vary ice
shelf basal melt parameterisation and geothermal heat flux, but this is beyond the scope of the present study.

**2.2.3 Spatial resolution sensitivity tests**

The base resolution of the ice sheet model is 32 km. The AMR allows the areas covered by ice to be refined once to 16 km,
which shows some improvement to the simulated ice streams, although the difference is only about 1.2 m yr$^{-1}$ on average over
the whole ice sheet (Figs. C2a and C2b). Additional sensitivity simulations were performed refining only the areas of ice
streaming up to 8 km and up to 4 km (Figs. C2c and C2d). These tests showed that after refining the entire ice sheet to 16km,
the difference in average ice velocity for any further refinement of the ice streams converges to zero (Fig. C3) and the pattern





of major ice stream features (Fig C2), the position of the marine margins and the ice volume across the NH ice sheets is not
significantly changed, except across the southern area of the Eurasian ice sheet (Fig. C4). However, computational costs are
quadrupled with each level of refinement.  Thus, we determine one level of refinement (16 km) to be sufficient for this study
in which we are focussing more on the large-scale geometry of the ice sheet rather than the finer details of the ice streams.
This is a similar conclusion to that drawn from the simulations presented by Albrecht et al., (2020) and Gandy et al., (2019),
the latter further showing anything finer than 4 km does not improve the match of simulated ice streams to empirical data.
There is an increase in the velocity of up to around 3000 m yr$^{-1}$ at the centre of some of the ice streams at the higher resolutions,
which could be important during simulations of the deglaciation (Robel and Tziperman, 2016). We performed an additional
simulation refining the ice streams across the marine section of the Eurasian ice sheet to 2 km to see if any marine processes
would be captured that could not have been resolved at lower resolutions. This did not lead to any significant difference in the
ice velocity in this region compared to the 4 km simulation (Fig. C2e), but again could be important in deglaciation simulations
when MISI could be triggered (Gandy et al., 2020; Patton et al., 2015; Petrini et al., 2020; van Aalderen et al., 2024).
**2.3 Experiment design**
**2.3.1 Boundary and initial conditions**
The coupled simulations broadly follow the PMIP4 protocols for the LGM (Kageyama et al., 2017) and the PGM (Menviel et
al., 2019), which prescribe greenhouse gases, orbital parameters and the Antarctic Ice Sheet configuration. Following the
method of Patterson et al., (2024), we also prescribe SSTs and Sea ice from HadCM3 simulations of 21 ka and 140 ka (Figs.
1a and 1b). A description of the HadCM3 simulations, the justification for this choice of approach, and a discussion on how
these SSTs may affect the result is also presented by Patterson et al., (2024). Vegetation is kept at pre-industrial distribution,
which could have an effect on the results since studies have shown the importance of the albedo-vegetation feedback during
glacials, particularly for the PGM (Colleoni et al., 2009b; Crucifix and Hewitt, 2005; Stone and Lunt, 2013; Willeit et al.,

317    2024).

The interactive ice sheet model domain covers the whole NH, including the North American, Greenland and Eurasian ice
sheets. Patterson et al., (2024) showed that the initial ice sheet model conditions used in the glacial maxima simulations
overwhelmingly determined the configurations of the final ice sheets due to the ice-albedo feedback, and that the climate at
the glacial maxima had an opposite impact on the difference in NAIS ice volume between the LGM and PGM to what was
expected. This suggests that the evolution of the climate and the ice sheets leading up to the glacial maximum are important
in determining the configurations of the ice sheets at the glacial maximum. We, therefore, chose to initialise the LGM and
PGM simulations from the respective ice sheet reconstructions available to ensure realistic ice sheet geometry for each period,
accounting for the evolution of the climate and ice sheets prior to the glacial maxima. With this approach, we can examine
how the differences in ice geometry and background climate between the two time periods affect the sensitivity to the model
parameters that control key earth system feedbacks (e.g. ice-albedo feedback, ice-elevation feedback and climate-ice sheet



interactions). The LGM orography was initiated from the GLAC-1D reconstruction (Briggs et al., 2014; Ivanovic et al., 2016;
Tarasov et al., 2012; Fig. 1c) and the PGM was initiated from a combination of a simulated PGM NAIS by Patterson et al.,
(2024) and simulated PGM EIS by Pollard et al., (2023) (Fig. 1d) and their corresponding topographies.

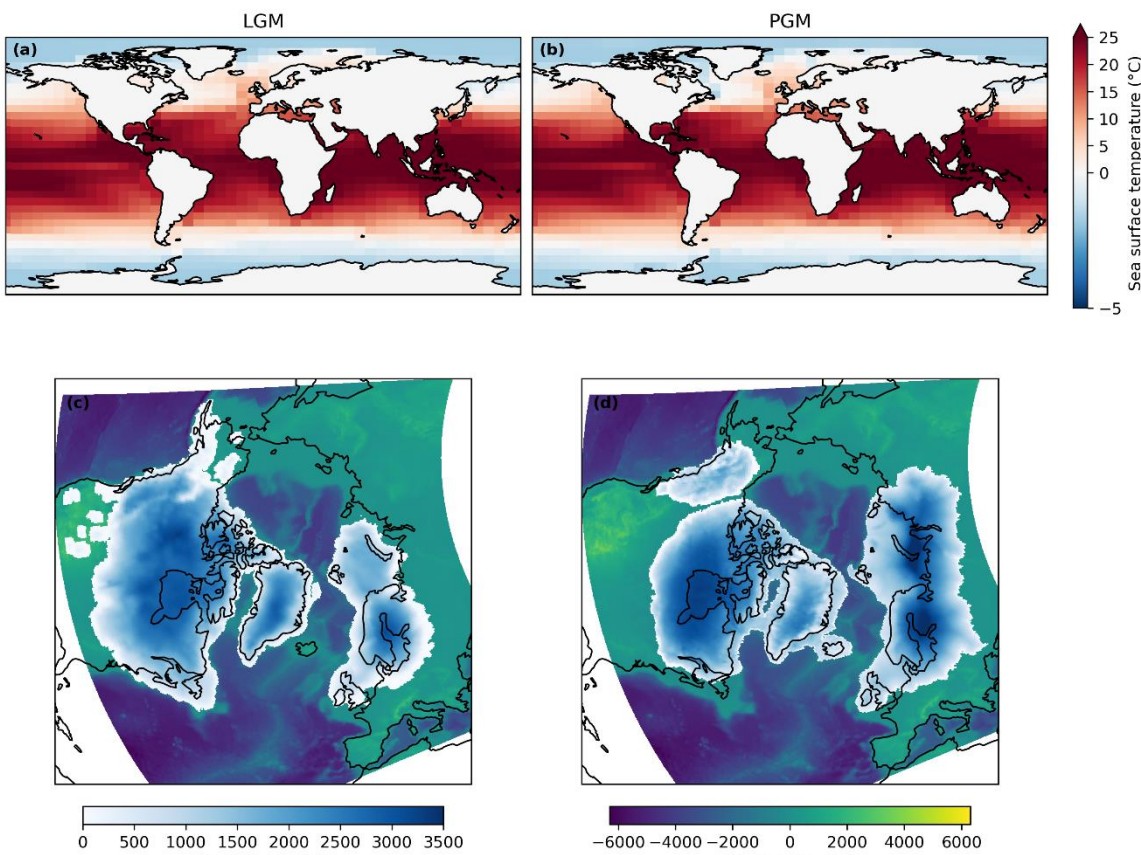


**Figure 1: Boundary and initial conditions for the LGM and PGM simulations. Sea surface temperatures prescribed in the FAMOUS atmosphere model for (a) LGM and (b) PGM; and initial topography (meters above sea level) and ice thickness in the BISICLES ice sheet model interactive domain for (c) LGM and (d) PGM.**

**2.3.2 Ensemble design**
As well as the initial ice sheet conditions, modelled ice sheet volumes and areas are also sensitive to a number of
parameterisations related to climate processes, surface mass balance and ice sheet dynamics. To assess this sensitivity, we
design an ensemble using maximin Latin Hypercube Sampling (Williamson, 2015; Santner et al., 2003), that consists of 120
combinations of 12 uncertain climate and ice sheet model parameters, varied over a specified range (Table 1). These 120
simulations are each run with the LGM and PGM initial conditions described in Sect. 2.3.1, resulting in 240 total simulations.
Each was integrated for 500 climate years (5000 ice sheet years). Since we start from a glacial maximum configuration and





spun-up internal temperatures, this is enough time for the ice sheets to (i) reach equilibrium (or close to it), and (ii) give an
indication of whether the parameters are producing reasonable ice sheets and form ice streams. Each simulation took around
35 hours running on 8 cores to complete (~280 core hours).
The choice and range of parameters is adapted from several previous ensemble studies (Gandy et al., 2023; Gregoire et al.,
2011; Patterson et al., 2024; Sherriff-Tadano et al., 2024). We vary three uncertain parameters related to ice sheet dynamics
in BISICLES; the basal friction coefficient in the power law relation (*beta*), the till water drain factor (*drain*), and the sub-
shelf melt constant (*c*). The *elevcon* parameter controls the magnitude of the height adjustment applied and the remaining
parameters control the climatic conditions and ice albedo in the simulations.

**Table 1: Parameters varied in the ensemble and the ranges sampled.**

| Parameter | Unit | Ensemble range | Notes |
|---|---|---|---|
| **Weertman friction coefficient, *beta*** | Pa m$^{-1/3}$ a$^{1/3}$ | 20,000 to 60,000 | Represents the resistance of ice at the base to motion. The higher the value, the stronger the friction between the ice and the bedrock over which it is flowing. |
| **Till water drain factor, *drain*** | yr$^{-1}$ | 0.01 to 0.05 | Controls the rate of vertical till-stored drainage and therefore water pressure in the till layer. The higher the value, the more rapidly till water is removed. |
| **Sub-shelf melt constant, *c*** | m yr$^{-1}$ °C$^{-1}$ | 1 to 50 | Characterises the relationship between ocean thermal forcing and sub-shelf melt rate |
| **Lapse rate, *tgrad*** | K m$^{-1}$ | -0.01 to -0.002 | Air temperature lapse rate used during downscaling to the ice sheet surface. The more negative the number, the stronger the lapse rate effects (Smith et al., 2021) |
| **Sensitivity of bare ice albedo, *daice*** | K$^{-1}$ | -0.4 to 0 | The sensitivity of bare ice albedo to surface air temperatures above the melt threshold (mimics darkening of the surface due to melt ponds forming in summer). The minimum value reduces the bare ice albedo to as low as 0.15 (Smith et al., 2021) |
| **Surface snow density threshold, *fsnow*** | kg m$^{-3}$ | 350 to 800 | The density threshold for snow beyond which the surface is regarded as bare ice. The higher the value, the higher the albedo for denser snow, tending to increase ice sheet albedo overall (Smith et al., 2021) |
| **Sensitivity to surface grain size, *av_gr*** | μm$^{-1}$ | 0 to 0.01 | The sensitivity of the surface snow albedo to increasing grain size. The higher the value, the more the albedo decreases over time, reducing snow albedo overall (Smith et al., 2021) |





| | | | |
|---|---|---|---|
| **Relative humidity threshold, *rhcrit*** | Pa$^{-1}$ | 0.6 to 0.9 | The threshold of relative humidity above which large-scale clouds form (Smith, 1990) |
| **Precipitating ice fall out speed, *vf1*** | m s$^{-1}$ | 1 to 2 | The precipitating ice fall out speed (Heymsfield, 1977) |
| **Cloud liquid water conversion rate, *ct*** | s$^{-1}$ | 5x10$^{-5}$ to 4x10$^{-4}$ | Rate of conversion of cloud liquid water droplets to precipitation (Smith, 1990) |
| **Cloud liquid water threshold, *cw*** | kg m$^{-3}$ | 1x10$^{-4}$ to 2x10$^{-3}$ | The threshold of cloud liquid water (over land) above which precipitation forms (Smith, 1990). |
| **Height correction, *elevcon*** | | 1 to 1.5 | Scaling factor for the height of the vertical levels read by the ice sheet model (this study) |


### 2.4 Evaluating the ensemble

To evaluate the performance of the LGM ensemble members and find sets of model parameters that produce NROY ice sheet configurations, we employ an implausibility metric. This allows a robust comparison of model output to empirical evidence and previous modelling studies, taking into account their uncertainties. The implausibility metric considers constraints on LGM ice volume, ice extent and Global Mean Air Temperature (GMT) derived from studies using palaeo-records of past climate and ice sheets and numerical modelling (Table 2). Since the PGM is poorly constrained in these areas, we are unable to evaluate the performance of the PGM ensemble in the same way. Instead, we opt to select the PGM ensemble members that correspond to the selected LGM members to enable comparison, see whether the same parameter values produce plausible PGM ice sheets based on known configuration differences and allow us to learn more about the PGM without the restriction of uncertain constraints.

The NAIS area is evaluated based on the southern extent of the ice sheet reconstructed by Dalton et al., (2020), within ± 3 times the area of the ice lobes (Fig 2a). We set this envelope of uncertainty (based on ice-lobe area) to account for known common model biases, such as over-estimated Alaskan ice, and limitations such as the inability to simulate the dynamic ice lobes (Patterson et al., 2024). Similarly, the plausible range of the EIS is considered to be within ± 3 times the area of the BIIS (Fig. 2b) based on the reconstruction from Hughes et al., (2016), since none of our simulations maintain ice over this area (see Sect. 3.1) and we do not want to compensate for/hide this limitation by over-estimating ice elsewhere. The GMT range is



determined from different estimated levels of LGM cooling, and their uncertainties, relative to a pre-industrial GMT of 13.7 ±
0.1 °C (1880-1900; NOAA National Centers for Environmental Information, 2023; Sherriff-Tadano et al., 2024).

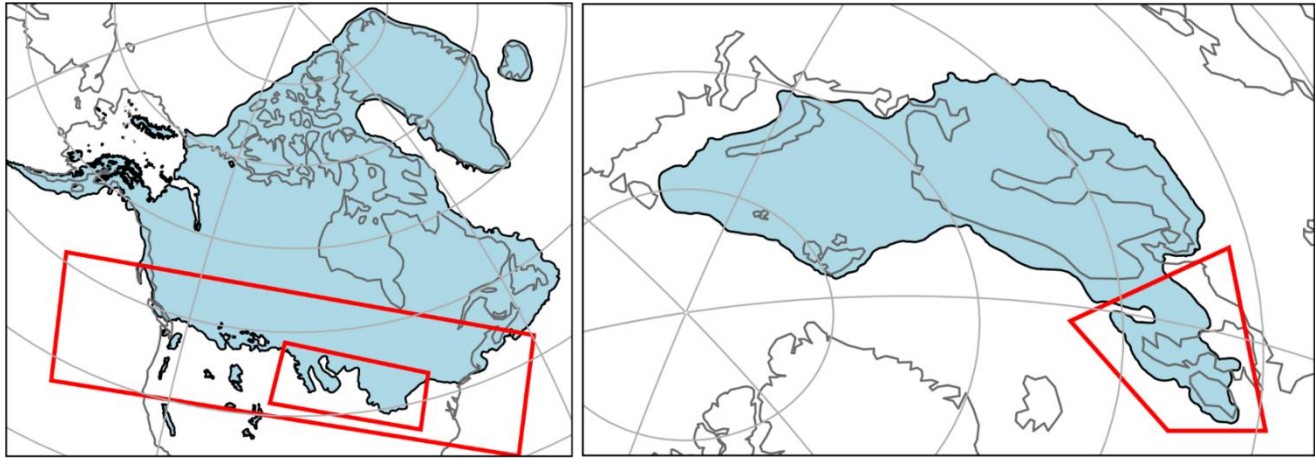


**Figure 2: Reconstructions used in the implausibility metric. (a) North American Ice sheet extent from Dalton et al., (2020)*; the large
red box delimits the southern extent footprint used in the implausibility metric; the smaller red box indicates the area of the lobes
used to calculate the range of plausible values. (b) Eurasian ice sheet extent from Hughes et al., (2016); the red box indicates the area
of the BIIS used to calculate the range of plausible ice areas.**

**Table 2: The ranges of plausible values for ice sheet volume and extent (expressed in metres global mean sea level equivalent; m sle),
and global mean surface air temperature (GMT; given in •C) used in our implausibility metric, and references to the published work
used to derive these ranges.**

| Metric | | Plausible range | References |
|---|---|---|---|
| **North American Ice Sheet (NAIS)** | Volume (m s.l.e.) | 68 – 88 | Abe-Ouchi et al., 2015; Gregoire et al., 2012; Lambeck et al., 2017; Moreno-Parada et al., 2023; Peltier et al., 2015; Simms et al., 2019; Tarasov et al., 2012 |
| | Area (km$^2$) | $2.0 \times 10^6 – 7.16 \times 10^6$ | Dalton et al., 2020 |
| **Eurasian Ice Sheet (EIS)** | Volume (m s.l.e.) | 13 – 23.5 | Abe-Ouchi et al., 2015; Hughes et al., 2016; Lambeck et al., 2006; Patton et al., 2016; Peltier et al., 2015; Tarasov et al., 2012 |
| | Area (km$^2$) | $3.83 \times 10^6 – 8.02 \times 10^6$ | Hughes et al., 2016 |
| **Global Mean surface air Temperature (GMT; °C)** | | 5.6 - 12.1 | Annan et al., 2022; Annan and Hargreaves, 2013; Holden et al., 2010; Liu et al., 2023; Osman et al., 2021; Schmittner et al., 2011; Schneider von Deimling et al., 2006; Zhu et al., 2022 |



## 2.5 Gaussian process emulation and Sobol sensitivity analysis

To determine which of the model parameters had the most influence on the uncertainty in modelled ice sheet configurations, and whether this differed for each of the NH ice sheets and each glacial maxima, we perform a Sobol Sensitivity Analysis (Saltelli, 2002; Sobol′, 2001) on four diagnostics for each ensemble; NAIS ice volume, NAIS southern area, EIS ice volume and EIS area. This produces a first order sensitivity index which measures the contribution to the output variance by each model parameter alone; a second order index which measures the contribution from interactions between two parameters and; a total order index which is the contribution by a model parameter as a result of its first order sensitivity and all higher order interactions. An index value of 0.05 is often used as the threshold above which a parameter is considered to have an important influence on the output variance (Zhang et al., 2015).

The Sobol analysis requires a uniform sample of thousands of model inputs, for example, generated following Saltelli's extension of the Sobol sequence, which are outside of our initial parameter sample. This would therefore require additional evaluations of the model, which would require significant additional computational resources. To this end, we train Gaussian Process (GP) emulators (Kennedy and O'Hagan, 2001; Oakley and O'Hagan, 2004) on each of the four diagnostics from the two 120 member ensembles. These emulators are then employed to evaluate the additional parameter sets generated by the Sobol sequence. Using this sequence and the emulators, we are able to generate and evaluate more than 200,000 samples in only a few minutes, a number which would have been computationally intractable using FAMOUS-BISICLES directly. Since we use a complex model with a large number of uncertain parameters, a sample of this size is necessary in order to increase the reliability of the Sobol analysis.

To evaluate the performance of our emulators and ensure their predicted output is sensible compared to the modelled output, we perform a Leave-One-Out Cross-Validation (LOOCV) on each emulator (Bastos and O'Hagan, 2009; Rougier et al., 2009). In general, leave-k-out cross-validation involves splitting the dataset of input parameters and output diagnostics into separate training sets and testing sets. The emulator is trained using the training set and then fed the input parameters of the testing set to evaluate. The values it then predicts can be compared to the actual modelled values. In the case of the LOOCV, all but one set of inputs and outputs are used as the training set and the emulator is used to predict the output left out. This process is then repeated for each of the 120 model outputs. We found that, compared to the modelled outputs, seven of the ensemble input parameter sets consistently produced poor predictions for four or more of the eight diagnostics. Therefore, to improve the quality of the emulator fit, we removed these seven inputs, re-trained the emulators, and once again performed the LOOCV. The predicted values (and their 95% credible intervals) compared to the modelled values for each emulator are shown in Appendix D (Fig D1). Overall, between 84-93% of the predicted intervals contain the true model output, which we determine is enough for the purposes of the Sobol analysis.





# 3 Results and discussion

## 3.1 Initial ensemble

After running the ensembles of simulations for the LGM and PGM, we obtain two sets of 120 simulations with a wide spread of NH ice sheet configurations (Fig. 3). The ensemble mean volume of the NAIS at the LGM is 37.6 m s.l.e., with a smaller mean at the PGM of 22.8 m s.l.e.. In contrast, the LGM has a smaller mean EIS volume of 5.39 m s.l.e. compared to 12.6 m s.l.e. at the PGM. Both ensembles have a similar mean Greenland ice sheet volume of ~ 7 m s.l.e.. The range in ice volume and extent across the ensembles are shown in Figs. 3 and 4 which reveal a larger spread in NAIS volume at the LGM but a larger EIS spread at the PGM. Figure 4a shows that the LGM simulations tended to have more extensive ice across the Laurentide ice sheet and in the area joining the Laurentide to the Cordilleran ice sheet, but that the PGM had more extensive ice to the south and east of the EIS and over Alaska while maintaining an ice free corridor between the Laurentide and Cordilleran. Whilst these relative volumes and extents between the LGM and PGM are consistent with knowledge of the different NH ice sheet configurations at each glacial maxima, the average values are much lower than current estimates suggest. This is due to a large proportion of the ensemble members deglaciating to very low or zero ice extent (Fig. 3).

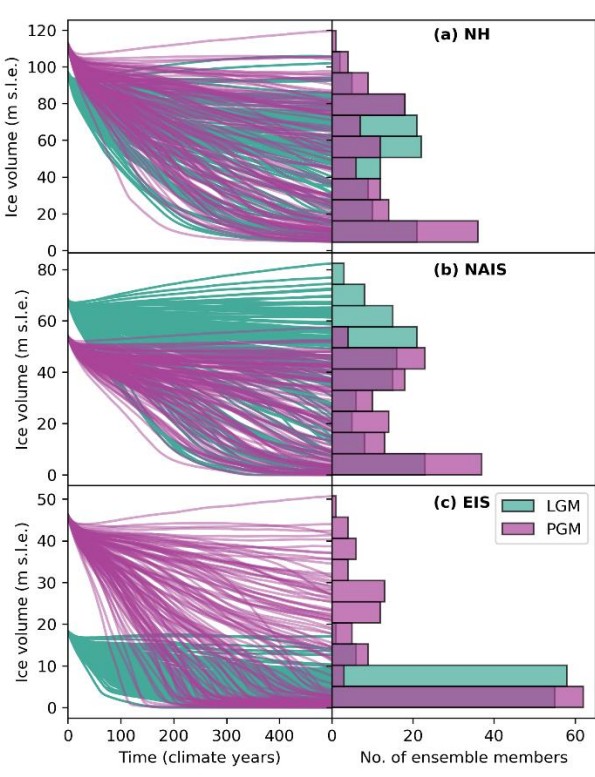

**Figure 3: Time series of ice volume over the 500 climate years (5000 ice sheet years) of simulation for each ensemble member (left hand panels) and histograms of the distribution of final ice volumes across the ensembles (right hand panels) for the LGM and PGM (a) Northern Hemisphere; (b) North American ice sheet and (c) Eurasian ice sheet.**





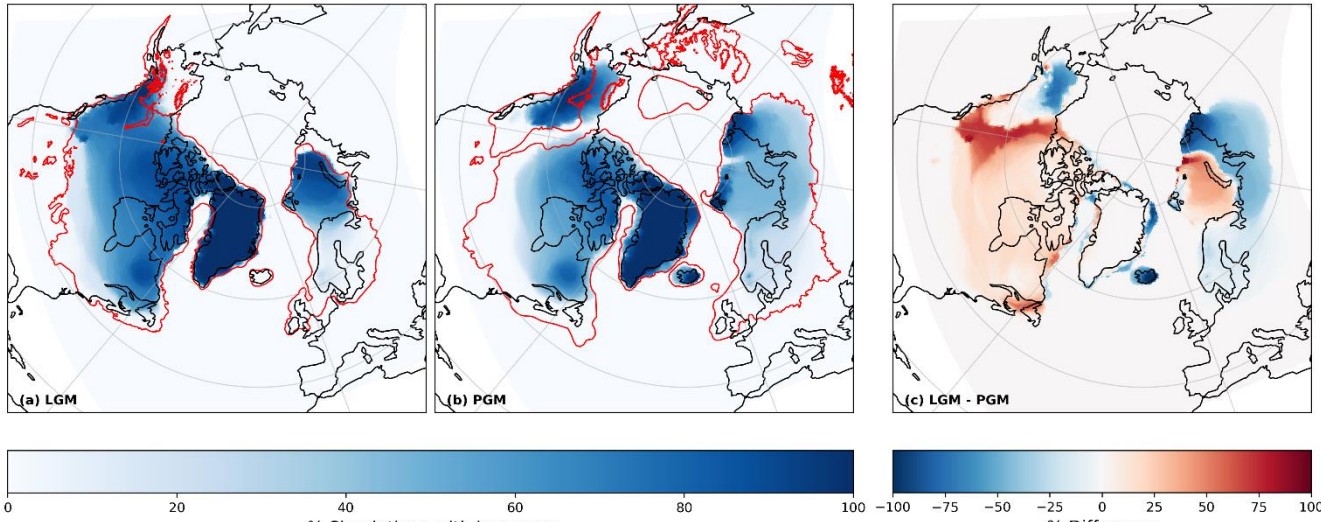

**Figure 4: Percentage of ensemble members that had ice over areas of the domain for (a) the LGM (with the extents of Dalton et al., (2020) and Hughes et al., (2016) in red); (b) the PGM (with Batchelor et al., (2019) extent in red); and (c) the difference between the LGM and PGM ensembles.**

In particular, all simulations lack a British-Irish Ice Sheet (BIIS) and most display a poor match to reconstructions over Scandinavia and in the southern margin and eastern marine extent of North America. This is due to large negative SMB values over these regions (Fig. 5) causing rapid deglaciation, with the BIIS disappearing in 600 ice sheet years or less. This is a similar result to Bradley et al., (2024) who used a GCM to simulate the SMB across the LGM ice sheets. Their simulations showed large ablation areas across the BIIS, the southern margin of Scandinavia and the southern, Pacific and Atlantic margins of the NAIS, but low melt rates across the Barents-Kara Ice Sheet and Greenland. Whilst they did not use a dynamical ice sheet model, they concluded that if this SMB pattern was applied to one, it would very likely drive rapid retreat of the southern margins of both ice sheets.



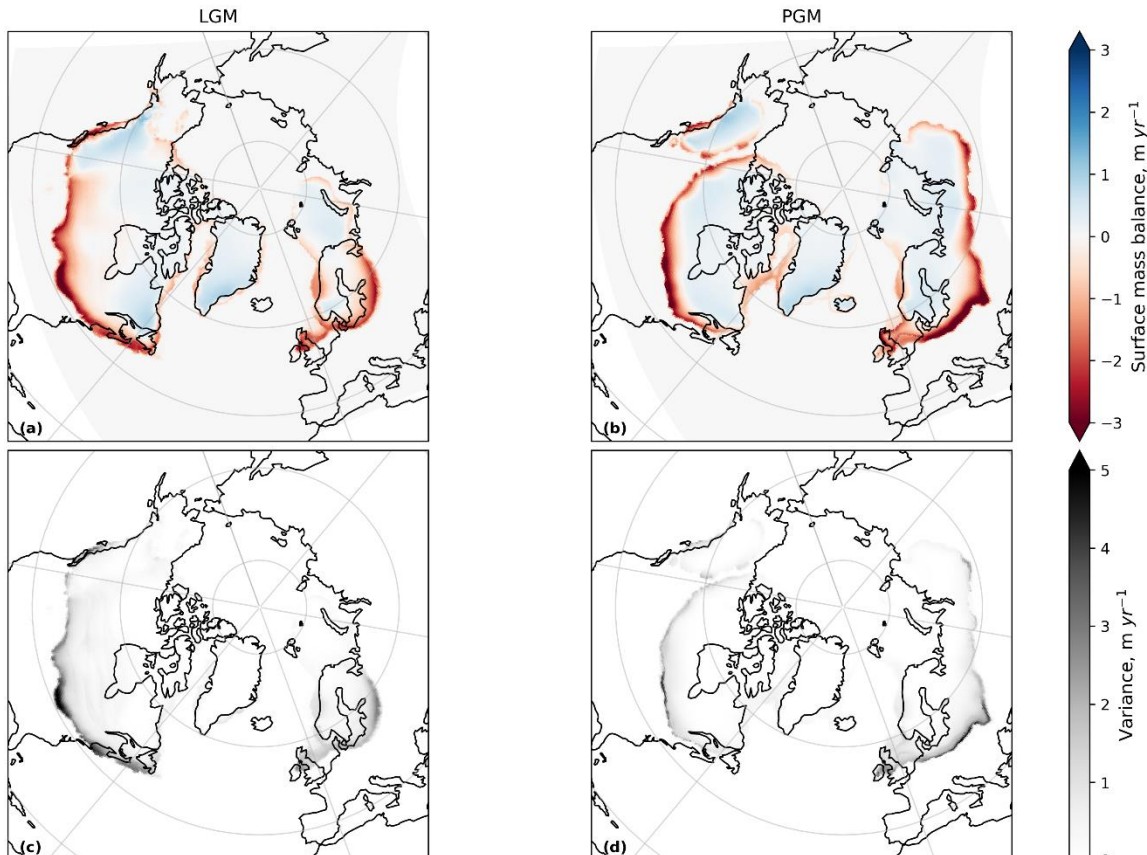

**Figure 5: Ensemble mean surface mass balance and variance at ice sheet year 200 for (a) and (c) the LGM and (b) and (d) the PGM.**

This result could reflect the asynchronous timing of the local maxima of the NH ice sheets since, for example, there is evidence that much of the NAIS reached its maximum extent at ~25 ka (Dalton et al., 2022, 2023) and the BIIS reached it maximum at ~25-23 ka before starting its retreat at ~22 ka due to a warming trend caused by a change in orbital parameters between 26–21 ka (Clark et al., 2022; Hughes et al., 2016). However, these reconstructions of the NAIS and BIIS still suggest there was extensive ice over these regions at 21 ka even if not at their maxima. In addition, Bradley et al., (2024) also performed a simulation using boundary conditions for 26 ka and obtained a similar result to 21 ka. They therefore concluded that the too negative SMBs are likely a result of biases in the simulated climate or ice sheet reconstruction, a highly non-equilibrated climate and ice sheet at the LGM, and/or the need to retune the model for LGM climate conditions (as also shown to be necessary by Gandy et al., 2023). Indeed, many other numerical modelling studies have also found it difficult to maintain extensive ice in these regions using a range of different models, boundary conditions and model parameters (van Aalderen et al., 2023; Quiquet et al., 2021; Scherrenberg et al., 2023b; Sherriff-Tadano et al., 2024; Ziemen et al., 2014; Zweck and Huybrechts, 2005).





In this present study, the compromise with using a coarse resolution model is that it is not able to accurately capture some of
the smaller scale atmospheric circulation effects that influence precipitation and temperature patterns. This leads to biases in
the modelled climate that result in some areas of the ice sheets not matching reconstructions. For example, simulations of the
NAIS have grown too much ice over Alaska and the southern extents are not extensive enough (Patterson et al., 2024; Sherriff-
Tadano et al., 2024; Ziemen et al., 2014). This is likely a result of an underestimation of the stationary wave effect on
temperature patterns; a common feature when using low resolution atmospheric models (Abe-Ouchi et al., 2007; Ganopolski
et al., 2010; Liakka et al., 2012; Roe and Lindzen, 2001).
**3.2 Non-implausible parameter sets**
We apply the implausibility metric described in Sect. 2.4 to the ensemble of LGM simulations to see if there are any sets of
model parameters that produce plausible ice sheets. All ensemble members have a GMT that falls within the range included
in the implausibility metric due to the control in surface conditions imposed by the prescribed SSTs. The LGM simulations
range from 6.34 – 9.20 °C and the PGM from 7.12 – 10.12 °C. This suggests that the SSTs used produce plausible LGM and
PGM climates, causing a warmer PGM compared to the LGM, which is also in agreement with palaeo reconstructions and
other dynamical models (Bintanja et al., 2005; Colleoni et al., 2016). However, due to ice extent and volume, only two LGM
simulations are NROY (labelled as NROYa and NROYb). Furthermore, we acknowledge the risk that our evaluation metric
may be too tightly constrained by uncertain palaeo reconstructions; ice sheet volume, in particular, is not well known. We
therefore also apply the extent and volume constraints separately to explore additional plausible ice sheet configurations,
especially since the volume constraint is still very uncertain and our minimum volume for the NAIS is less lenient than limits
that have been used previously (e.g. Gandy et al., 2023; Sherriff-Tadano et al., 2024). This results in the selection of two more
ensemble members; one that meets only the ice extent criteria (labelled as NROY extent) and one that meets only the ice
volume criteria (labelled as NROY volume). All four of these NROY simulations are shown in Fig. 6, with the corresponding
four PGM simulations shown in Fig. 7. Time series of ice volume, surface mass balance, sub-shelf melt plus calving rate and
surface air temperature for these simulations can be found in Appendix E.
The final volumes and extents of the NROY simulations are outlined in Table 3. Overall, the LGM NROY simulations show
a good match to the reconstructed extents of the LGM ice sheets and the equivalent PGM simulations display a smaller NAIS
and larger EIS in line with empirical evidence and previous studies. Whilst the equivalent PGM simulations show a smaller
NAIS than the extent of Batchelor et al., (2019), this reconstruction represents the maximum MIS 6 extent (190-132 ka) and
therefore is likely larger than the 140 ka ice sheet would have been, particularly for the NAIS. These four NROY model
simulations suggest the NAIS was ~25 m s.l.e. smaller at the PGM compared to the LGM, and the EIS ~24-27 m s.l.e. larger.
There are very few existing reconstructions of the PGM ice sheets and none produced using a coupled climate-ice sheet model.
Our simulations perform well in comparison to these reconstructions (Fig. 8) and thus provide a great alternative for use as
boundary conditions in future climate and sea level modelling studies.



**Table 3: Ice sheet volumes and extents at the end of the 5000 ice sheet years for the two NROY LGM simulations and the corresponding PGM simulations**

| | LGM | | | | PGM | | | |
|---|---|---|---|---|---|---|---|---|
| | NROYa | NROYb | NROY extent | NROY volume | NROYa | NROYb | NROY extent | NROY volume |
| **NAIS Volume (m s.l.e.)** | 72.6 | 76.9 | 64.7 | 82.4 | 48.1 | 52.2 | 41.5 | 57.5 |
| **EIS Volume (m s.l.e.)** | 14.2 | 17.0 | 12.7 | 13.7 | 38.7 | 44.0 | 35.6 | 50.7 |
| **NAIS area (southern area) ($\times 10^6$ km$^2$)** | 14.2 (4.44) | 13.9 (4.17) | 12.4 (2.91) | 13.1 (3.51) | 10.9 (1.87) | 10.8 (1.66) | 9.31 (0.75) | 10.1 (1.32) |
| **EIS area ($\times 10^6$ km$^2$)** | 4.53 | 5.0 | 4.08 | 3.56 | 9.86 | 10.1 | 9.04 | 9.61 |

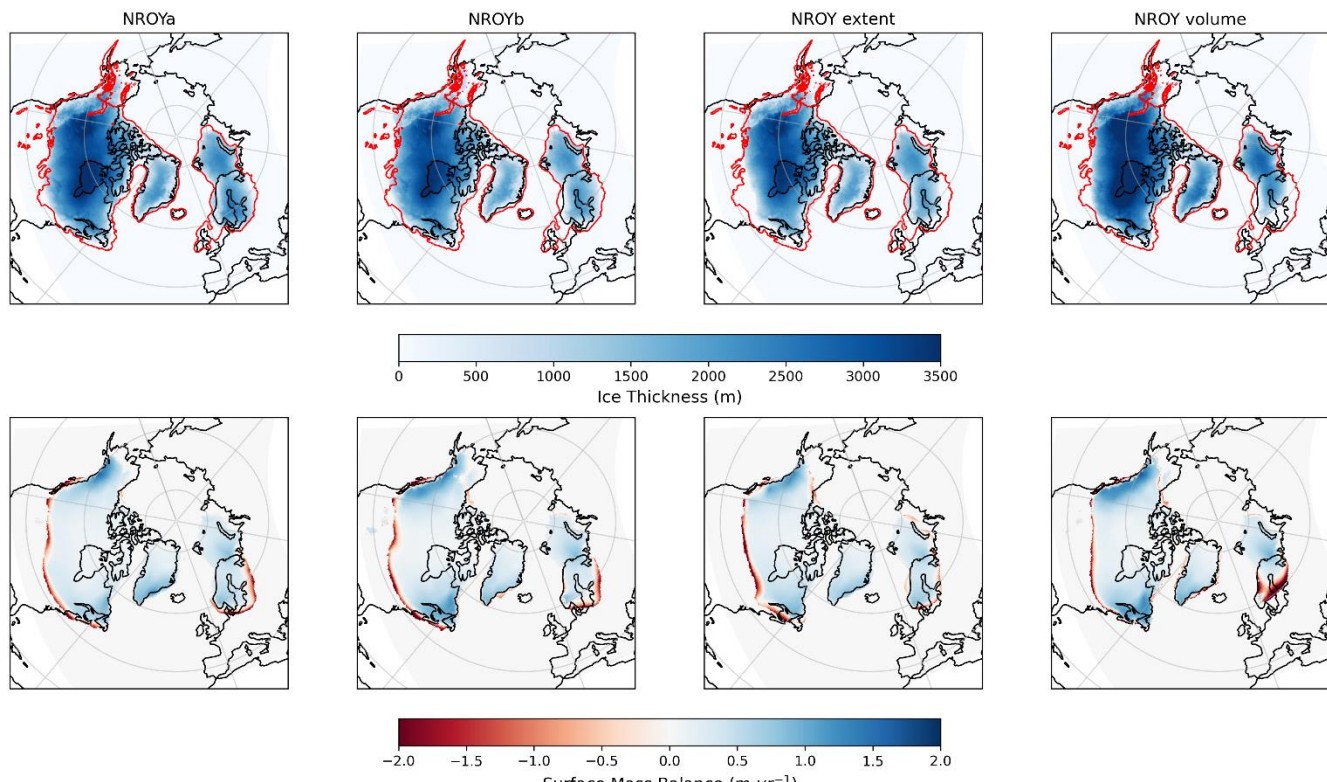

**Figure 6: Final ice thickness and surface mass balance for the four NROY LGM simulations**

All NROY simulations still lack a BIIS, however, which suggests that biases in the climate model are the cause rather than model parameter values. Due to high rates of sub-shelf melt ($\sim$ 60-75 m yr$^{-1}$), the NROY simulations also lack ice shelves by



the end of the 5000 ice sheet years, which could also have contributed to the underestimation of the eastern margin of the
NAIS and the deglaciation of the BIIS (Scherrenberg et al., 2023b). However, there are not many constraints on the extent of
ice shelves during the LGM or PGM since they leave few glaciological traces behind. There is some evidence that a large,
thick ice shelf extended into the Arctic Ocean during the MIS 6 glaciation (Jakobsson et al., 2016; Svendsen et al., 2004) and
during the last glaciation a thick ice shelf may have covered Baffin Bay (Couette et al., 2022). Similarly, the rate of sub-shelf
melt is poorly constrained during past periods, however, since some studies have shown ocean driven melt to be important for
the evolution of the marine based sectors of the NH ice sheets (Alvarez-Solas et al., 2019; Clark et al., 2020; Petrini et al.,
2020), it may be useful to implement a more complex parameterisation or perform some additional sensitivity tests to explore
this process further in future studies.

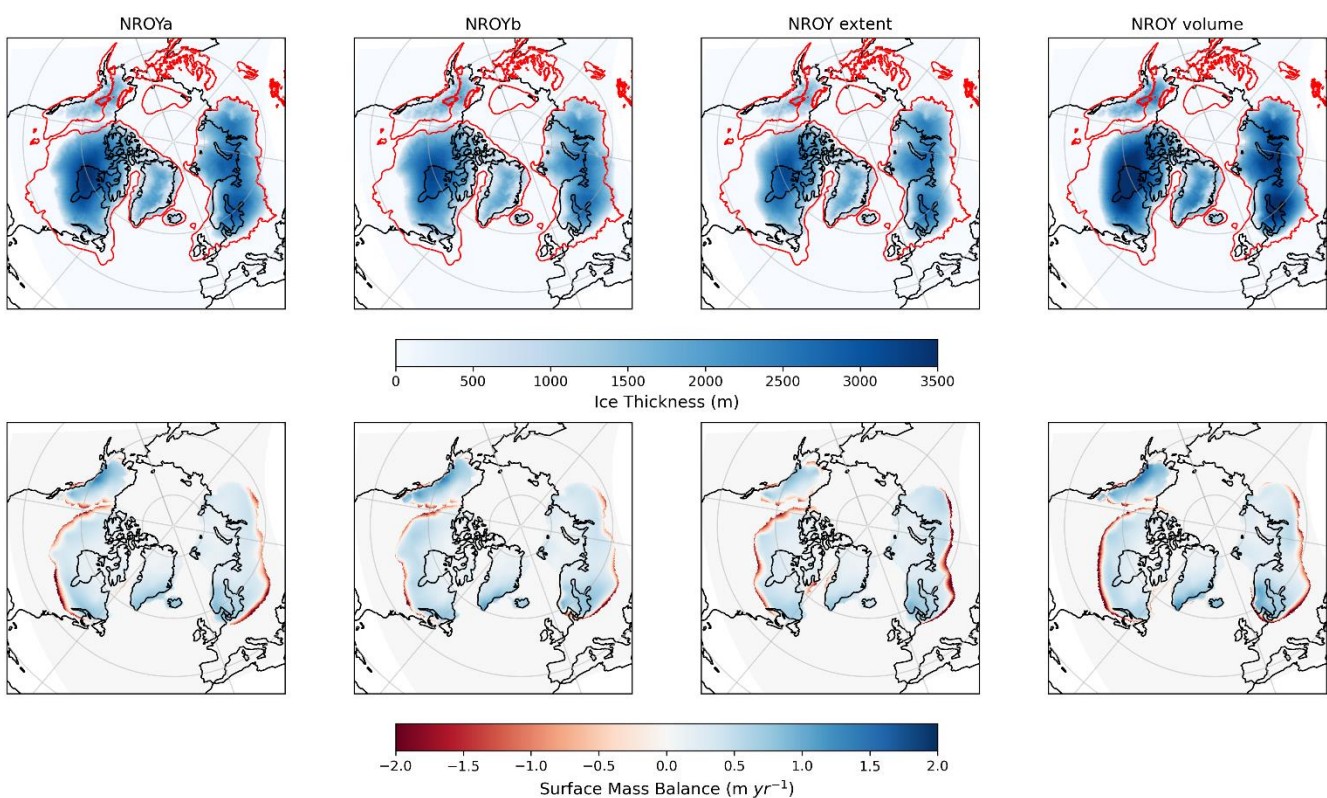


**Figure 7: Final ice thickness and surface mass balance for the four NROY PGM simulations**
Despite difficulties in the past in obtaining a sufficient southern extent of the NAIS in lower resolution models, the NROYa
and NROYb simulations do a relatively good job, only falling short of the Dalton et al., (2020) reconstruction by 3 % and 9
%, respectively. The two additional NROY simulations are less close to the reconstructed extent, however, and all four still
fail to capture the ice lobe structures. This is because they are formed by extensions of terrestrial ice streams as a result of
complex ice dynamics and subglacial processes (Jennings, 2006; Margold et al., 2018). They are also highly asynchronous,
dynamic features resulting in their glacial maximum limits being very uncertain (Dalton et al., 2020; Margold et al., 2018).





Therefore, it is not surprising that a relatively low resolution climate and ice sheet model with highly idealised subglacial
environments is unable to resolve such features (Gandy et al., 2019; Zweck and Huybrechts, 2005).

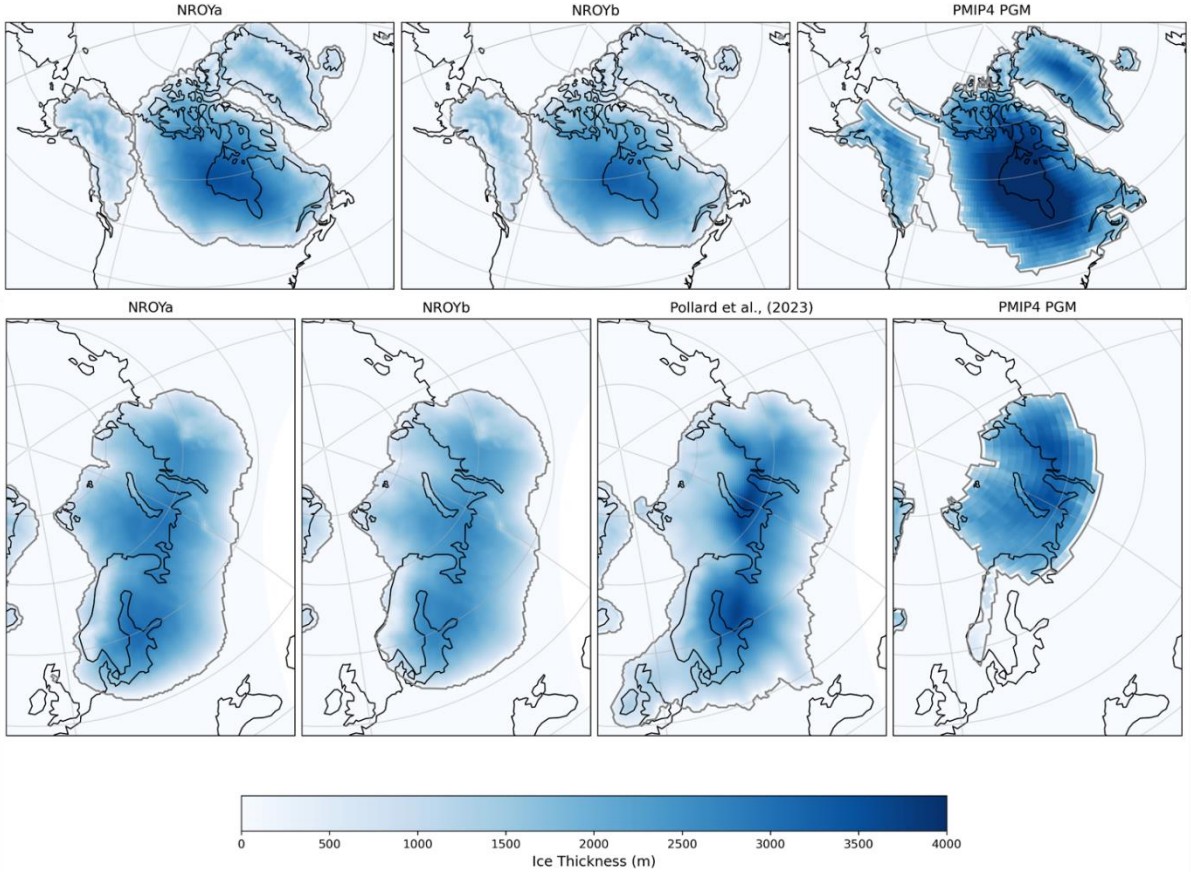

**Figure 8: Comparison of the two NROY PGM simulations to other model reconstructions (Abe-Ouchi et al., 2013; Pollard et al., 2023)**

The parameter values used in the two NROYa and NROYb simulations are in similar areas of the parameter space for all
parameters except *tgrad* and *drain,* suggesting the ice sheets are fairly insensitive to these two parameters (Supplementary Fig.
S1). Interestingly, Figs. 9a and 9b show that, if considering the NAIS and EIS separately, there are five simulations that produce
only a plausible NAIS but do not meet constraints for the EIS. Furthermore, as we have already seen, there are also simulations
that produce plausible ice sheet extents but fall short on the volume and vice versa. Many of these simulations are situated in
different areas of the parameters space than the two NROY simulations for most of the parameters (Supplementary Fig. S1).
Figures 9c and 9d show that the NROYa and NROYb parameter sets also produce the largest PGM ice sheet extents in the
ensemble but there are additional simulations that produce similar or larger volume ice sheets, which, in relation to the EIS,
was not the case for the LGM. These results all suggest that both ice sheets and both time periods display different sensitivities
to model parameters.



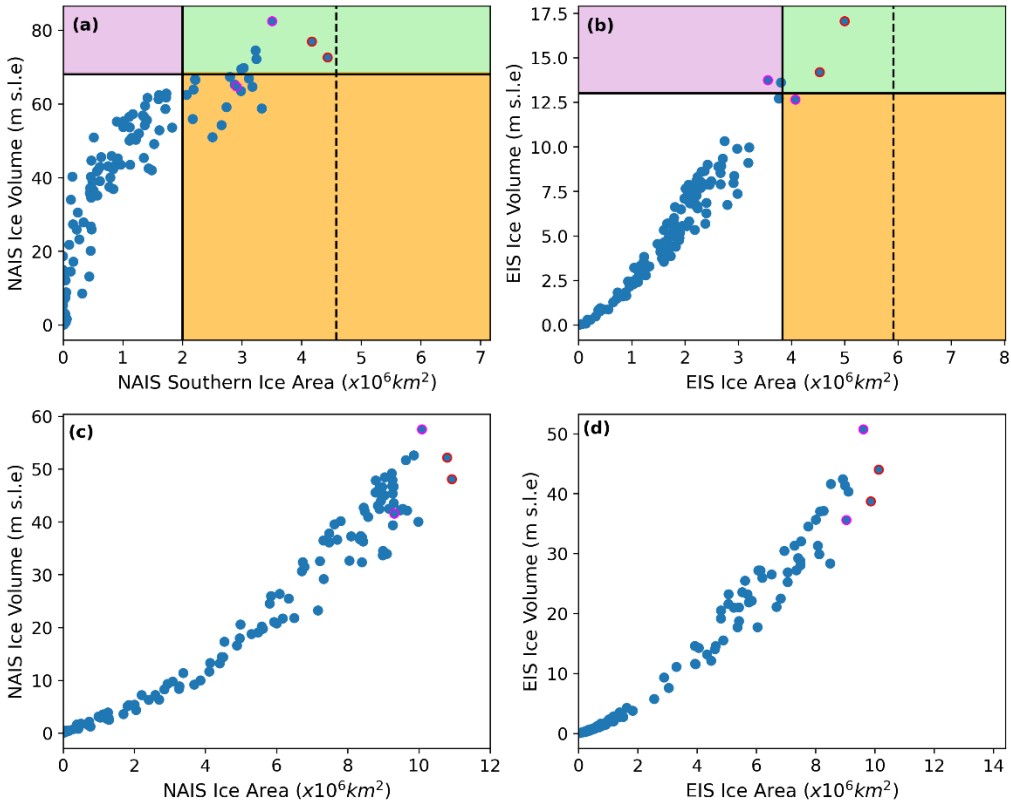

**Figure 9: Results from the full ensembles of simulations showing (a) LGM North American ice sheet southern area versus volume and (b) LGM Eurasian ice sheet area versus volume. The solid lines show the minimum values used in the implausibility metric for area and extent and the dotted line shows the actual extent of the ice sheet reconstructions. Simulations that fall within the green box satisfy area and volume constraints for each individual ice sheet, the orange box indicates they satisfy the area constraints only and purple only the volume constraints. The points outlined in red are the two NROY simulations (i.e. fall into the green box for both ice sheets) and the points outlined in pink are the additional NROY extent and NROY volume simulations. Panels (c) and (d) show the equivalent results for the PGM ensembles without the constraints.**

## 3.3 Sensitivity to parameters

To examine and quantify these different sensitivities we perform the Gaussian Process emulation and Sobol Sensitivity analysis described in Sect. 2.5. Due to the performance of the emulators leading to some uncertainty in the predicted values and therefore the values of the Sobol indices, we are careful to not over interpret the results and only analyse the highest values and largest differences. We also use emulation to isolate the relationship between certain influential parameters and ice sheet volume in which the emulator predicts the model output across a sample of the range of one parameter whilst all other parameters are held at their midpoint values.

The first and second order sensitivity indices for the NAIS and EIS volumes for the LGM and PGM are shown in Fig. 10a and 10b and the difference in sensitivities between the two ice sheets in Fig. 10c. The analysis indicates that the ice sheets were relatively insensitive to the parameters *vf1*, *drain*, *ct*, *rhcrit* and *c*. The insensitivity to the value of the sub-shelf melt is



unsurprising despite previous studies reporting a high sensitivity of the Antarctic and Eurasian ice sheets (Alvarez-Solas et al.,
2019; Berdahl et al., 2023; Berends et al., 2023). This is because the simulations lost their ice shelves fairly soon into the
model run due to either high rates of sub-shelf melt resulting from the large values of *c*, or large ablation rates as a result of
other climate model parameter values.
The most influential parameters in all aspects are *fsnow* and *av_gr*, which control the albedo of the ice sheet, with larger values
of *fsnow* and smaller values of *av_gr* leading to larger ice sheets. The third albedo parameter, *daice*, is also important,
particularly for the NAIS, having a positive correlation with ice sheet size. However, as in the case of NROY extent, the value
of *daice* is less important provided that *fsnow* is high and *av_gr* is low since these produce a high enough albedo to maintain
an extensive ice sheet on their own (Fig. F1). These three parameters also have important interactions with other parameters
and each other. This importance of the albedo parameters is consistent with previous studies investigating the sensitivity of the
NAIS to uncertain parameters (Gandy et al., 2023; Patterson et al., 2024; Sherriff-Tadano et al., 2024), but our detailed Sobol
sensitivity analysis is able to not only identify the most important parameters but also quantify the importance of all the other
parameters. Furthermore, the inclusion of the EIS in our analysis reveals the importance of some other parameters for the
configuration of the EIS. This includes *beta*, *cw* at the LGM, and, despite the value of *tgrad* being in different areas of the
parameter space for the NROY simulations, this analysis shows that the EIS is highly sensitive to this parameter, especially
for the PGM. The NAIS is also sensitive to new parameters introduced in this study that weren't tested in Gandy et al., (2023)
or Patterson et al., (2024). This includes *beta*, and for the LGM the volume is also impacted by the value of *elevcon*.
Here we discuss some of the possible reasons these four parameters (*elevcon, cw, tgrad* and *beta*) could have an effect on the
various ice sheets. However, further simulations and testing would need to be carried out to come to any conclusions. One
reason that the LGM NAIS shows a particular sensitivity to **elevcon** could be related to the size of the ice sheets since it affects
higher ice elevations more and, indeed, the value of the Sobol index for this parameter is in line with the average thickness of
each ice sheet. The fact that a larger value of *elevcon* leads to a larger NAIS (Fig. 11a) but doesn't impact the size of the EIS
could explain why the ensemble produced more plausible North American ice sheets at the LGM but did not perform as well
for the Eurasian ice sheet (Fig. 9). It may also explain some of the difference in NAIS size between the LGM and PGM.
Similarly, the LGM EIS being more sensitive to the value of **cw** than the NAIS or either PGM ice sheet could explain why
there are more simulations that produced larger volume Eurasian ice sheets at the PGM than the LGM, but the NAIS behaved
similarly between both periods (Fig. 9). *Cw* has a positive correlation with EIS volume up to a value of around 0.0012 kg m$^{-3}$
(Fig. 11b). Any increase above this does not appear to increase the ice volume much further. This could be because lower
values of *cw* cause increased precipitation due to decreasing the threshold of cloud liquid water above which precipitation
forms. This has a particular effect in summer leading to higher rainfall rates over the Northern Hemisphere continents which
contributes to the surface melting of the ice sheets through the flux of heat from the rain to the ice. One reason the LGM EIS
is particularly susceptible to this effect could be due to its smaller size. Precipitation is not downscaled onto elevation tiles in
the coupling, rather the coarse atmospheric output is applied to the ice sheet model which leads to rainfall being spread across
relatively large areas of the ice sheet, therefore affecting a large proportion of the LGM EIS (Smith et al., 2021). Another



reason could be related to the change in liquid cloud cover and its effect on the energy balance. The increased precipitation
leads to a decrease in the fraction of cloud cover which would allow a higher receipt of incoming shortwave radiation, thus
increasing the surface melt. However, the downwelling longwave radiation may also be decreased which would have the
opposite effect, decreasing the absorbed energy. Since the accumulation zone usually has a high albedo, reflecting much of the
incoming solar radiation, the SMB of this area is mostly controlled by changes in the longwave fluxes. In contrast, the low
albedo ablation zone is largely impacted by the shortwave radiation budget in the summer melt season. This latter process has
been found to be dominant in studies of the Greenland Ice Sheet, with reduced cloudiness contributing to its mass loss and
increasing its sensitivity to warming (Hofer et al., 2017; Izeboud et al., 2020; Mostue et al., 2024; Ryan et al., 2022). Again,
due to its smaller size, a large proportion of the LGM EIS is under ablation (54 % compared to around 35 % for the other ice
sheets in Fig. 5), potentially explaining why it is so sensitive to changes in cloud cover.

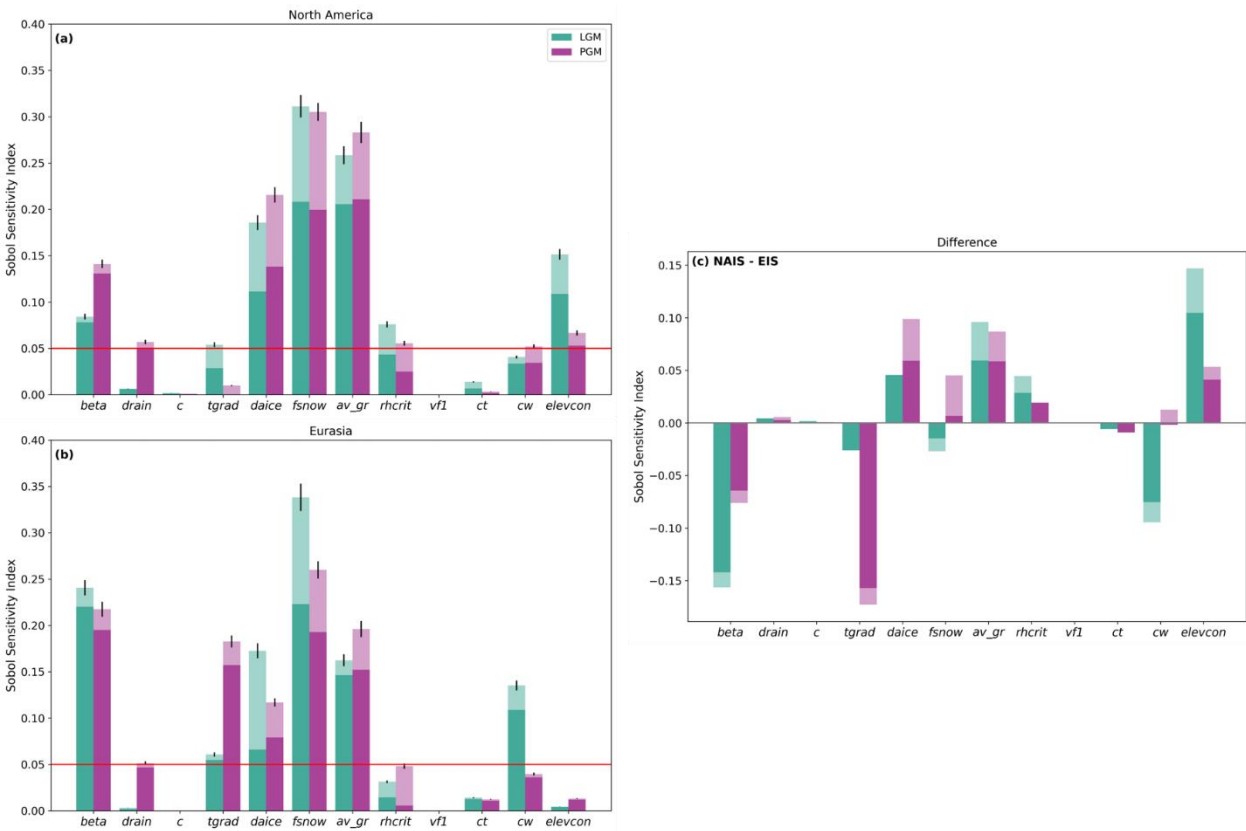


**Figure 10: The Sobol sensitivity index of the ice volume for each parameter for (a) the North American Ice Sheet and (b) the Eurasian**
**Ice Sheet. (c) The difference in sensitivity indices between the North American and Eurasian ice sheets. The darker colour represents**
**the first order index and the lighter colour the second order index (together showing the total sensitivity). The variance of the Sobol**
**indices plus the mean emulator variance is indicated by the black error bars. The red line indicates the index value of 0.05, above**
**which the sensitivity is significant.**
PGM EIS is much more sensitive to the value of *tgrad* than the other ice sheets. More negative values of *tgrad* cause a stronger
temperature-elevation feedback, resulting in warmer temperatures at lower elevations. This is going to have the largest impact





on ice sheets with larger ablation areas. Many of the simulated PGM Eurasian ice sheets collapse (Fig. 3) as a result of the
larger ice sheet being more unstable due to the larger GIA feedback. Therefore, many of these simulations will have strong
ablation over the Eurasian ice sheet that increases throughout the run, making it more sensitive to *tgrad* and the temperature-
elevation feedback.

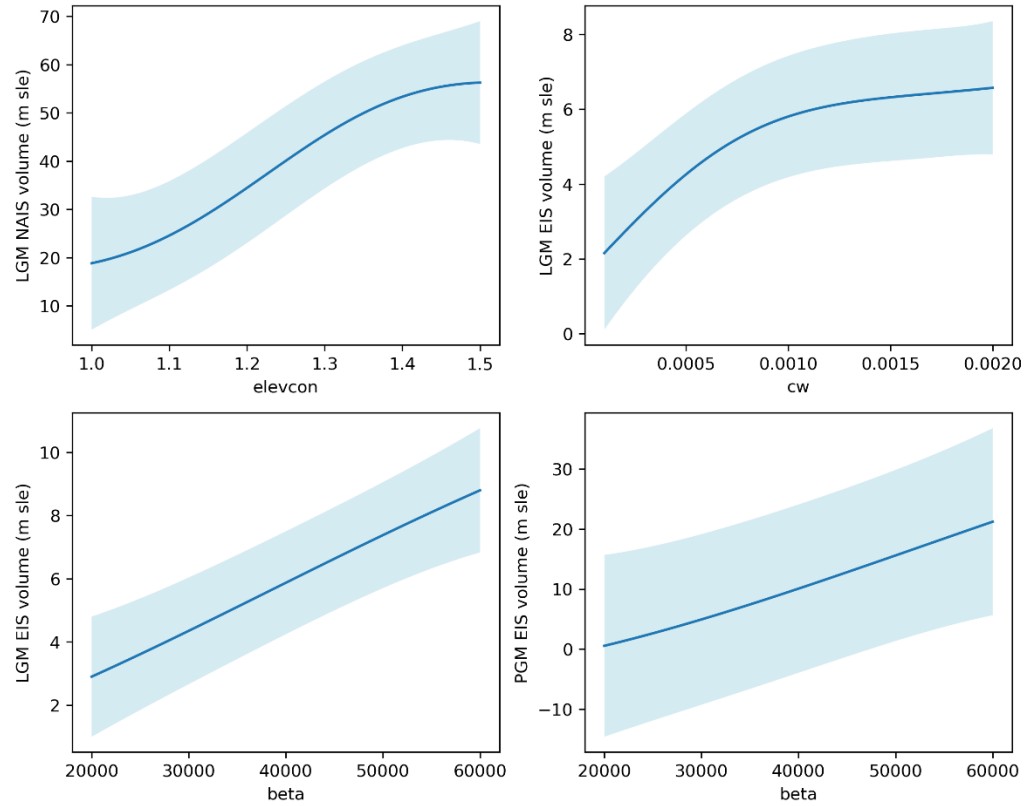


**Figure 11: The relationship between emulated mean ice sheet volumes and (a)** *elevcon* **, (b)** *cw* **, (c) and (d)** *beta* **a The 95th percentiles**
**are shown by the blue shaded region.**
In addition, **beta** has a positive correlation to the size of the Eurasian ice sheet at both the LGM and PGM (Fig. 11c and 11d)
but does not have as much of an impact on the NAIS which could also explain some of the different behaviours seen between
both ice sheets. *Beta* is also the only parameter that causes a large difference in the sensitivity indexes of volume and extent,
with the ice volume being much more sensitive (Fig. 12a). This could explain why the NROY extent simulation falls short of
the volume constraints since it has a relatively low beta value (Fig. F1). This also supports the idea that reduced basal friction
results in more ice mass loss from the Eurasian ice sheet compared to North America since faster flow from the interior of the
ice sheet to the more extensive marine margins causes a larger discharge of ice across the grounding line where it is calved or
lost by sub-shelf melting (Fig. 12b and 12c). This therefore affects the volume and thickness of the ice sheet but not so much
the extent since ice already reaches the edge of the continental shelf (Blasco et al., 2021; Scherrenberg et al., 2023a; Sherriff-
Tadano et al., 2024). Scherrenberg et al., (2023a) and Quiquet et al., (2021) show a similar impact of basal friction on ice sheet



volume compared to extent at the LGM but also show that the thinner ice sheets, larger ablation area and increased ice
velocities, caused by lower basal friction led to a faster deglaciation. Interestingly, both of the NROYa and NROYb simulations
have lower values of *beta* than the five additional simulations that produce a plausible NAIS but not EIS. This suggests that
the right combination of parameters, especially in regard to the albedo parameters *fsnow*, *av_gr* and *daice*, and the interactions
between parameters, can compensate for the faster flow and are thus more important for the size of Eurasia (Fig. F1).

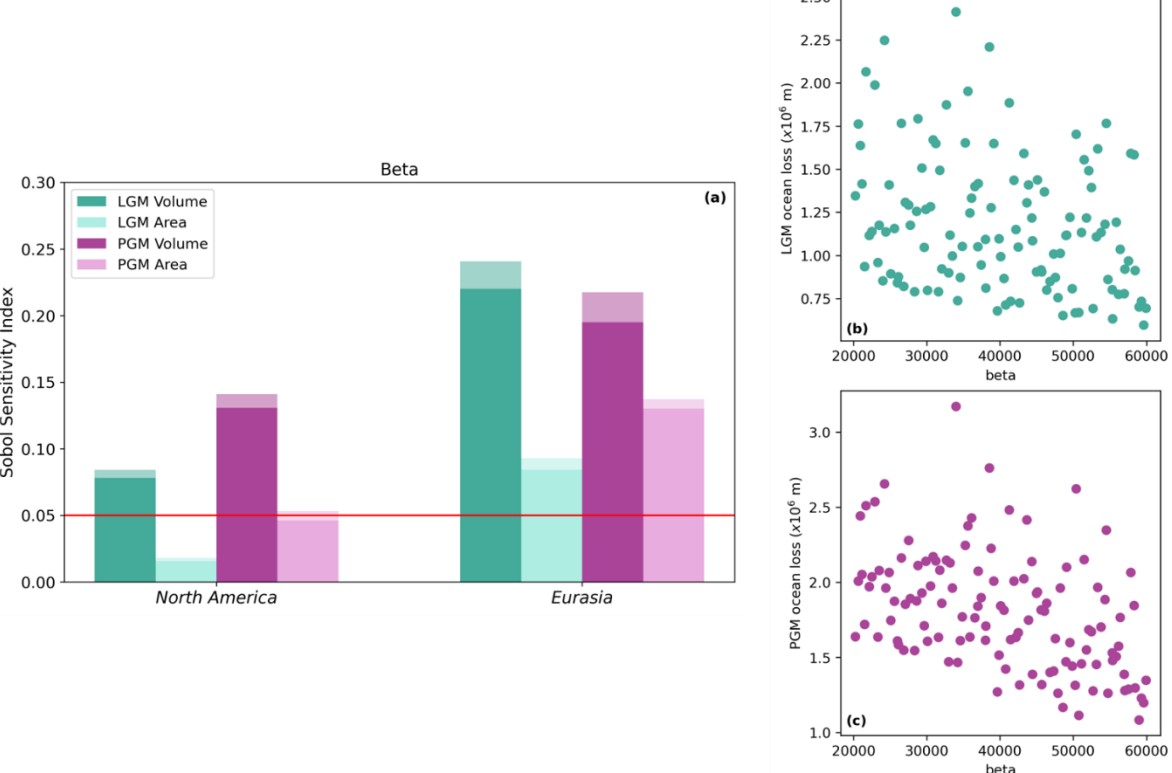


**Figure 12: (a) Sobol Sensitivity Indices for the ice volume and extent at the LGM and PGM for the parameter *beta* and (b) LGM**
**and (c) PGM total ice loss to the ocean (calving + sub-shelf melt) versus the value of *beta*.**

## 3.4 Ice dynamics

The representation of ice streams in the simulations was updated from the previous FAMOUS-BISICLES simulations of the
NAIS (Sherriff-Tadano et al., 2024) by performing the sensitivity tests and internal temperature spin up detailed in Sect. 2.2.
The velocity of areas of ice streaming in the NROY simulations range from a few hundred m yr$^{-1}$ to 5000 m yr$^{-1}$ which is a
similar range to what has been observed on present day Antarctica and Greenland (Joughin et al., 2010; Rignot et al., 2011).
We asses to what extent the modelled ice streams in the NROYa and NROYb simulations match empirical reconstructions by
performing a qualitative comparison to LGM reconstructions of the Laurentide ice streams (Fig.13a; Margold et al., 2018) and
the Eurasian ice streams (Fig.14a; Patton et al., 2017). For the Laurentide Ice Sheet, the locations of many of the ice streams
show good agreement, particularly in NROYb (Fig. 13b and 13c). Using the numbers and names used in Margold et al., (2018)



this includes; (1) Mackenzie Trough, (18) Amundsen Gulf, (123) Massey Sound, (129) Prince Gustaf Adolf Sea, (126) Smith
Sound/Nares Strait, (22) Lancaster Sound, (23) Cumberland Sound, (24) Hudson Strait, (45) Notre Dame Channel, (133)
Placentia Bay-Halibut Channel, (25) Laurentian Channel, (131) The Gully and (134) Northeast Channel IS. There are also
areas of general streaming where many smaller ice streams are found (numbers 108-116 and 167-170). One major ice stream
that is not very active in these simulations is (19) M'Clure Strait and there is a poor representation of ice streaming along the
southern margin of the Laurentide Ice Sheet.

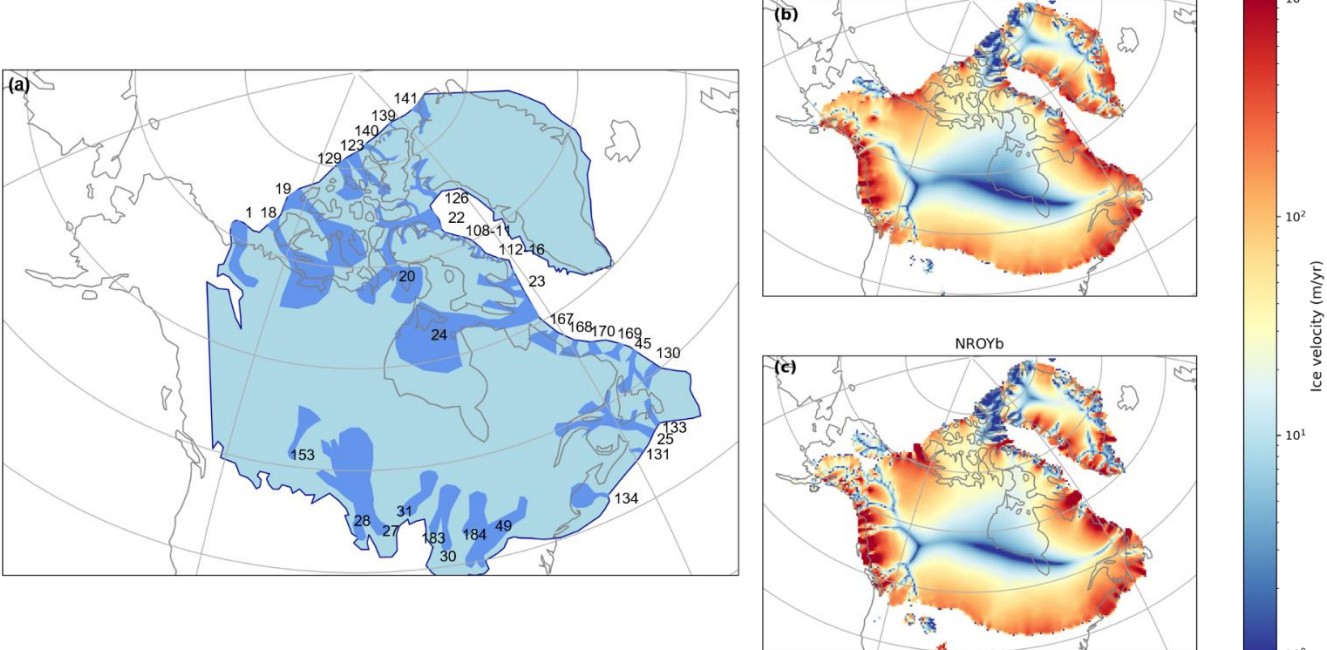

**Figure 13: (a) Empirical reconstruction of the active LGM Laurentide ice sheet ice streams (adapted from Margold et al., (2018),
and (b) NROYa and (c) NROYb ice velocities at the end of the 5000 year simulations.**

The Eurasian Ice Sheet does not have as defined areas of ice streaming, nevertheless, some of the major ice stream features
can be picked out (Fig. 14b and 14c). The following numbers relate to those in Fig. 14a and names are taken from van Aalderen
et al., (2023) and Stokes and Clark, (2001). There is some streaming activity in the location of one of the major ice streams;
(1) Bjornoyrenna ice stream and (10) Svyataya Anna ice stream is relatively well represented. Some of the smaller ice streams
are also modelled including; (2) Mid Norwegian, (8), (9), (11) and (12). However, other major and minor ice streams are not
active in these simulations; (3) Norwegian Channel, (4) and (5) Baltic Sea, (6) Gulf of Bothnia and (7). In addition, since the
BIIS is not present, neither are the ice streams in this region. Interestingly, there are active areas of ice streaming to the south
of the Barents Sea that are not present in the reconstruction. This could be due to the formation of a pro-glacial lake in this
region allowing the formation of ice shelves which have zero basal friction and therefore increase ice velocity (Sutherland et
al., 2020).



There are no comparable reconstructions of PGM ice streaming due to difficulties in dating and the erasure of glaciological
evidence following the Last Glacial advance. However, due to extent and topographic constraints on ice streaming, it is likely
that ice stream location was similar across the marine margins of the ice sheets (Pollard et al., 2023). The simulated PGM
NAIS velocity behaves similarly to the LGM but there is a lack of (1) Mackenzie Trough and a less pronounced (18) Amundsen
Gulf as a result of the different configuration of the ice sheets in this area (i.e. the location of the ice free corridor between the
Laurentide and Cordilleran ice sheets). However, there is more evidence of (19) M'Clure Strait in NROYa and more activity
on the southern Laurentide margin (Fig. G1). The PGM EIS velocity shows a more defined (3) Norwegian Channel ice stream
and NROYb has a better representation of (10) Svyataya Anna, (11) and (1) Bjornoyrenna ice stream than the LGM. There is
still no streaming in the Baltic Sea but the PGM also shows activity in the South Barents Sea. There is also additional ice
streaming in the Northeast where the PGM ice sheet extent further then at the LGM (Fig. G2).
Whilst the value of *drain* does not affect the volume or area of the ice sheets (Sect. 3.3) it has a significant effect on the ice
streaming/velocity of the simulations. The two NROY simulations display very different levels of ice streaming despite having
similar configurations largely as a result of having different values of *drain*. NROYa has a higher value of 0.04 causing
relatively quick drainage of the till water compared to NROYb which has a value of 0.01. Therefore, NROYb allows more
sliding since the effective pressure is lower and thus so is the basal shear stress. The value of *drain* may become more important
in simulations of deglaciations as ice streaming affects the stability of ice sheets and rate of retreat.




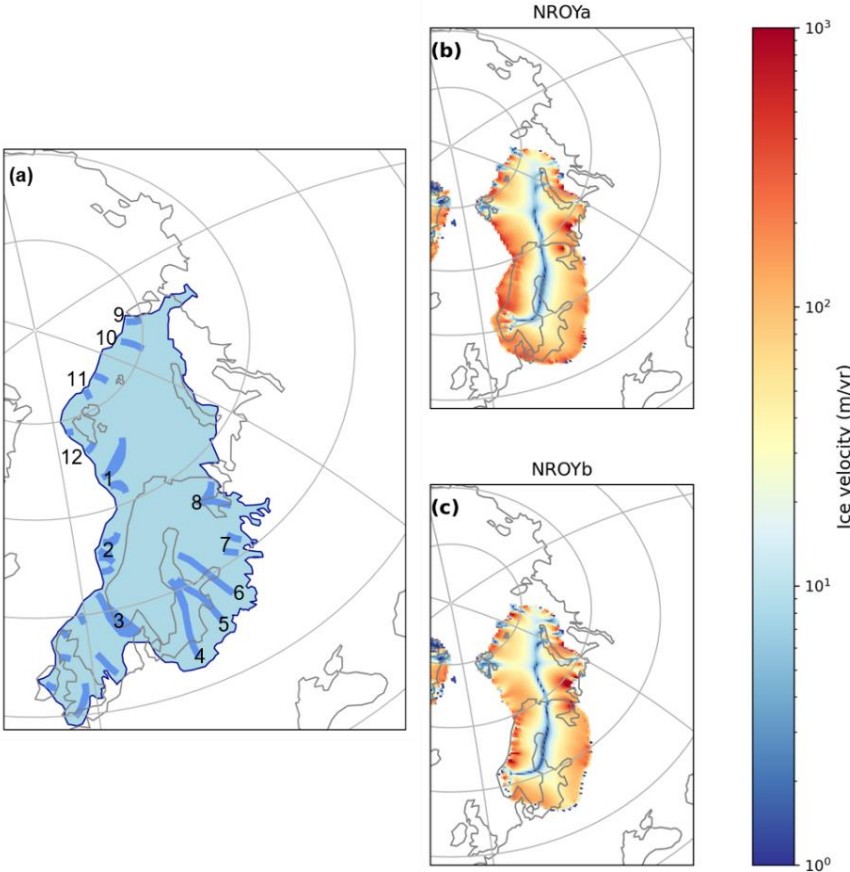

**Figure 14: (a) Empirical reconstruction of the location of active LGM Eurasian ice sheet ice streams (adapted from Patton et al., (2017), and (b) and (c) ice velocities at the end of the 5000 year NROY simulations.**

## 4 Conclusions

We ran ensembles of simulations using a coupled atmosphere-ice sheet model under LGM and PGM boundary conditions, varying uncertain climate and ice sheet model parameters. The model simulates plausible Northern Hemisphere ice sheets compared to empirical reconstructions and previous modelling studies, capturing the different configurations between the LGM and PGM. Through Gaussian Process emulation and a Sobol sensitivity analysis, we find that the volume and extent of both the simulated Northern Hemisphere ice sheets are sensitive to the parameters that control their albedo. However, the North American ice sheet and the Eurasian ice sheet, and the two glacial maxima, display different sensitivities to certain other parameters. The size of the North American ice sheet at the LGM is sensitive to the value of the height correction parameter (*elevcon*), the size of the Eurasian ice sheet is sensitive to the value of the lapse rate parameter (*tgrad*) at the PGM and to the basal friction parameter (*beta*) at both glacial maxima. This result highlights that, as well as the use of different initial conditions for the LGM and PGM, the difference in final ice volume and extent between both periods may also be impacted



by the choice of parameter values. However, after applying an implausibility metric we find two sets of NROY parameter
values that are plausible for both periods and both ice sheets, and we highlight an additional two simulations that we deem
NROY depending on the criteria used. We also do some work to improve the representation of ice streaming in the glacial ice
sheets and find that our simulations produce a good match to empirical reconstructions of LGM ice streams, especially in
simulations with lower values of till water drainage rate (*drain*).
The four NROY simulations produced in this study provide a good starting point for simulating and comparing the Last and
the Penultimate deglaciations, which will be the focus of future work. However, since it has been shown in the past that models
can be overtuned to certain climate conditions, it is not guaranteed that these parameter values will be conducive to the
deglaciation of the ice sheets in line with empirical reconstructions and work will need to be done to test this and calibrate the
model for both past and present conditions which will likely involve the use of emulators. In addition, there are some factors
that were not considered or not well represented in this work that may become more important for the deglaciation. These
include; the ice shelf melt parameterisation (Berends et al., 2023), the resolution at the grounding line (Gandy et al., 2021) and
the representation of proglacial lakes (Sutherland et al., 2020). This study was also limited by the use of prescribed surface
ocean conditions and pre-industrial vegetation and the absence of dust, all of which have been shown to initiate important
feedbacks for ice sheet evolution (Ganopolski et al., 2010; Obase et al., 2021; Willeit et al., 2024). Current modelling
capabilities prevented the use of a fully coupled atmosphere-ocean-ice sheet model with dynamic vegetation and dust for the
large number of simulations run in this study, however as technological advances are made to enable this in the future, running
similar simulations will provide useful information of the role of these other feedbacks on the evolution of the LGM and PGM
ice sheets.
**Appendices**
**Appendix A: Implementation of the *elevcon* parameter**
*elevcon* affects the surface temperature and SMB during the height adjustment to ice sheet tiles in the following manner;
• The effective elevation of each tile is multiplied by the value of *elevcon*. A value of 1.10 (10 %) means that the
704         elevation of an 1800 m tile has been increased to 1980 m.

• Surface air temperatures and longwave radiation are downscaled to each increased elevation tile.
• Surface fluxes and SMB are calculated based on the downscaled variables and other variables from the original
707         FAMOUS grid.

• The SMB and fluxes are then passed to the ice sheet and atmospheric models, but taken to represent the original tile
709         elevation, not the increased elevation to which the surface temperature was actually downscaled. For example, the
710         surface air temperature and SMB could be calculated on a 1980 m elevation tile, but they will be passed to the ice
711         sheet and atmospheric models as outputs from an 1800 m elevation tile.



Therefore, the increase in the tile elevation is only accounted for during the downscaling of surface temperature but is not
reflected when passing it to the ice sheet model or elsewhere in FAMOUS. In this way, additional cooling is applied over the
ice sheet interior by *elevcon*, which can be regarded as elevation-dependent height adjustment over ice sheets. This crudely
mimics the effect of the stable boundary layer in maintaining the cold surface condition in that area.
Two types of sensitivity experiments are performed with FAMOUS-BISICLES to validate the effect of *elevcon* on the modern
and LGM ice sheets and climates. The first sensitivity experiment is conducted under modern climate and the Greenland ice
sheet based on a control simulation performed by Lang et al. (in prep) and focuses on the effect of *elevcon* on the SMB. As
shown in Smith et al., (2021), the model simulates a mean ELA of approximately 1.8 km over the Greenland ice sheet, whereas
high resolution regional atmospheric models (e.g. MAR; Fettweis et al., 2013) suggest 1.2 km, meaning that the model
overestimates the ELA by 50 % (Fig. A1). Here, we applied an *elevcon* value of 50% and rerun the simulation. The inclusion
of the *elevcon* adjustment strongly suppresses the negative SMB seen around the elevation of 1 km to 2 km, and the ELA drops
from 1.8 km to approximately 900 m height (Fig. A1). Given that the ELA is now underestimated compared with the high-
resolution models, the value of 50 % appears to be too large and can be regarded as the upper limit. However, this sensitivity
experiment clarifies the substantial effect of *elevcon* on the SMB at the interior of the ice sheet. It further shows that *elevcon*
can be used to explore the effect of uncertainties in the SMB at the interior of the ice sheet arising from underestimating the
role of the stable boundary layer.
The second type of sensitivity experiments are performed under the LGM climate for the North American ice sheet. Here,
values of 10 %, 20 % and 50 % are tested with one of the ensemble members from Sherriff-Tadano et al., (2024) that exhibits
a strong local melting of the ice sheet from parts of the interior. Results are shown in Fig. A2. The strong local melting observed
around the Hudson Bay region in the control simulation is removed in all the sensitivity experiments. Also, depending on the
magnitude of the value of *elevcon*, the negative SMB seen at the eastern part of the Rocky Mountains is reduced and pushes
the ELA southwards.





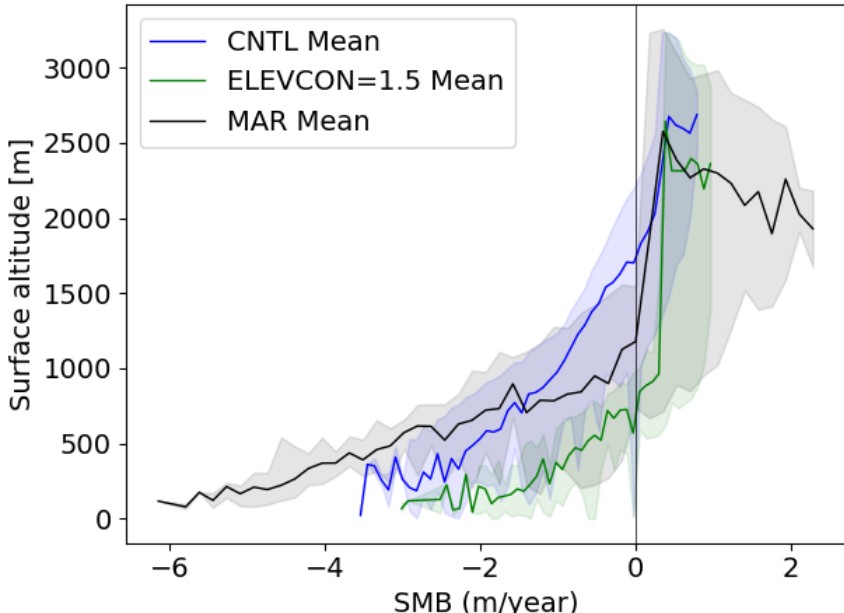


**Figure A1: Relation of SMB and surface altitude over the Greenland ice sheet in the modern climate simulations with FAMOUS-BISICLES. The blue line (shading) shows the mean result (range) from the control experiments, and the green shows those from the sensitivity experiments that include *elevcon* with a value of 1.5 (50 %). Also shown in black are the results from simulations using the MAR regional climate model (Fettweis et al., 2013).**

739

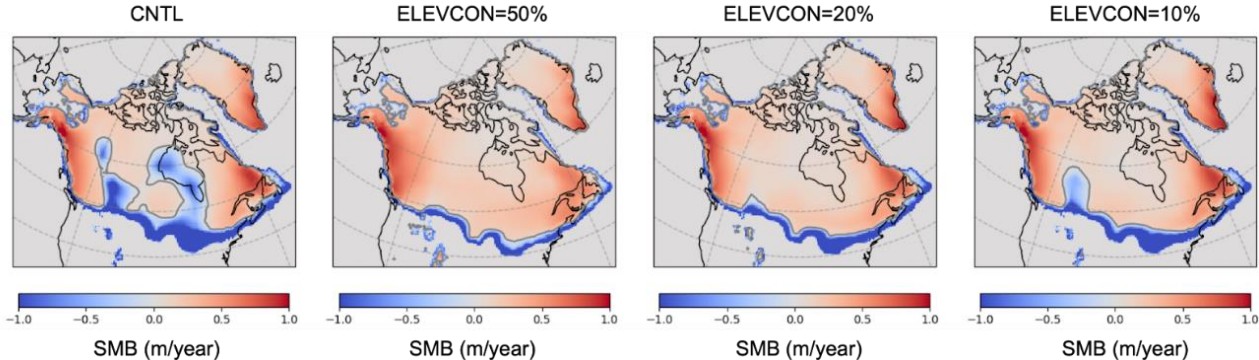

740

**Figure A2: Effects of different magnitudes of *elevcon* on the spatial pattern of SMB over the North American ice sheet at the LGM. CNTL corresponds to one of the ensemble members (xppma) in Sherriff-Tadano et al. (2024).**





## Appendix B: BISICLES spin-up

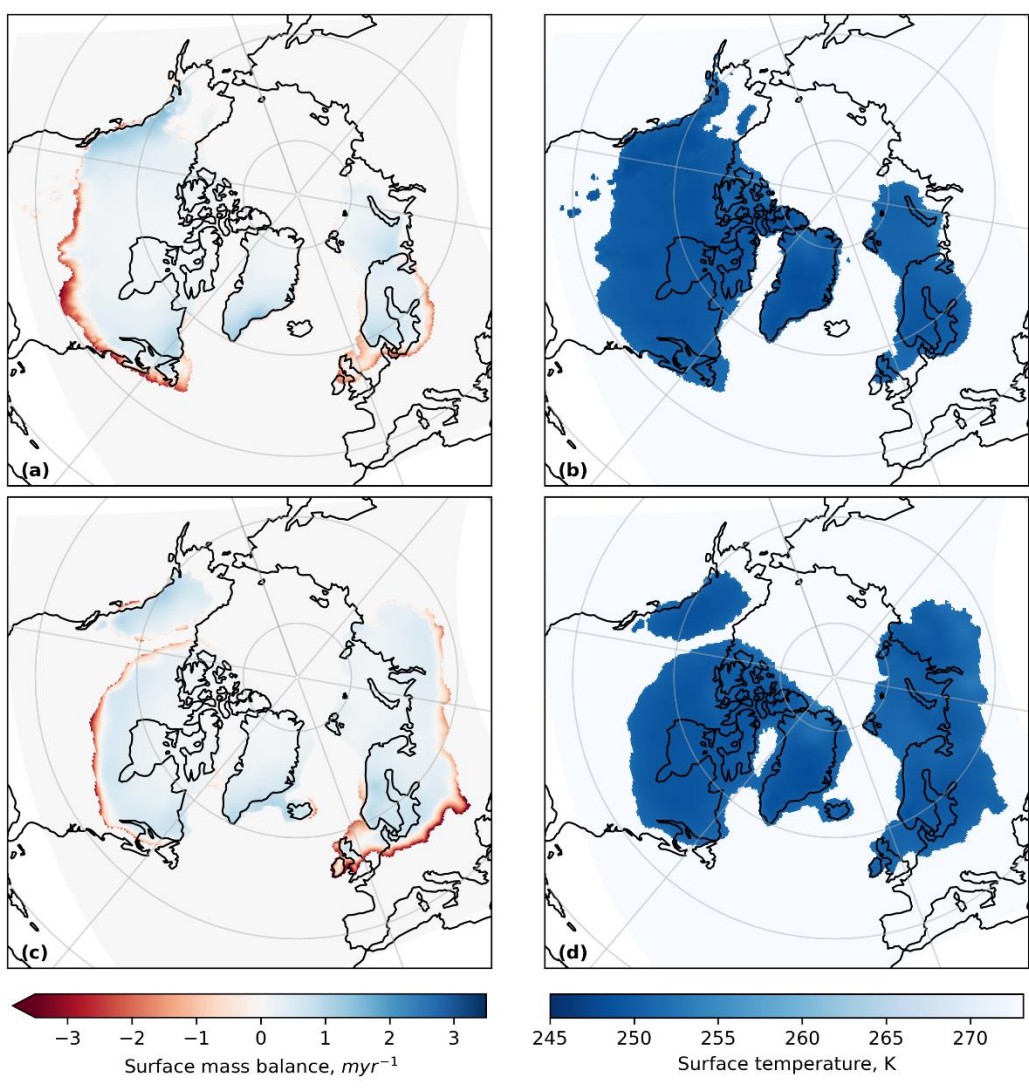

**Figure B1: Surface mass balance and ice surface temperature fields used in the (a), (b) LGM and (c), (d) PGM spin ups.**





746

**Figure B2: Cross section of LGM ice temperature at the end of the 20,000 year spin-up for the transects indicated by the red lines in (a), for the Euraisian ice sheet (b) and the North American ice sheet (c).**





**Figure B3: Cross section of PGM ice temperature at the end of the 20,000 year spin up for the transects indicated by the red lines in (a), for the Euraisian ice sheet (b) and the North American ice sheet (c).**



## Appendix C: Sensitivity tests



**Figure C1: Ice velocity after 5000 ice sheet years in simulations using till water drainage rates of (a) 0.005 m yr⁻¹, (b) 0.0199 m yr⁻¹, (c) 0.05 m yr⁻¹ and (d) 0.06 m yr ⁻¹.  All other parameters and initial conditions were kept the same.**



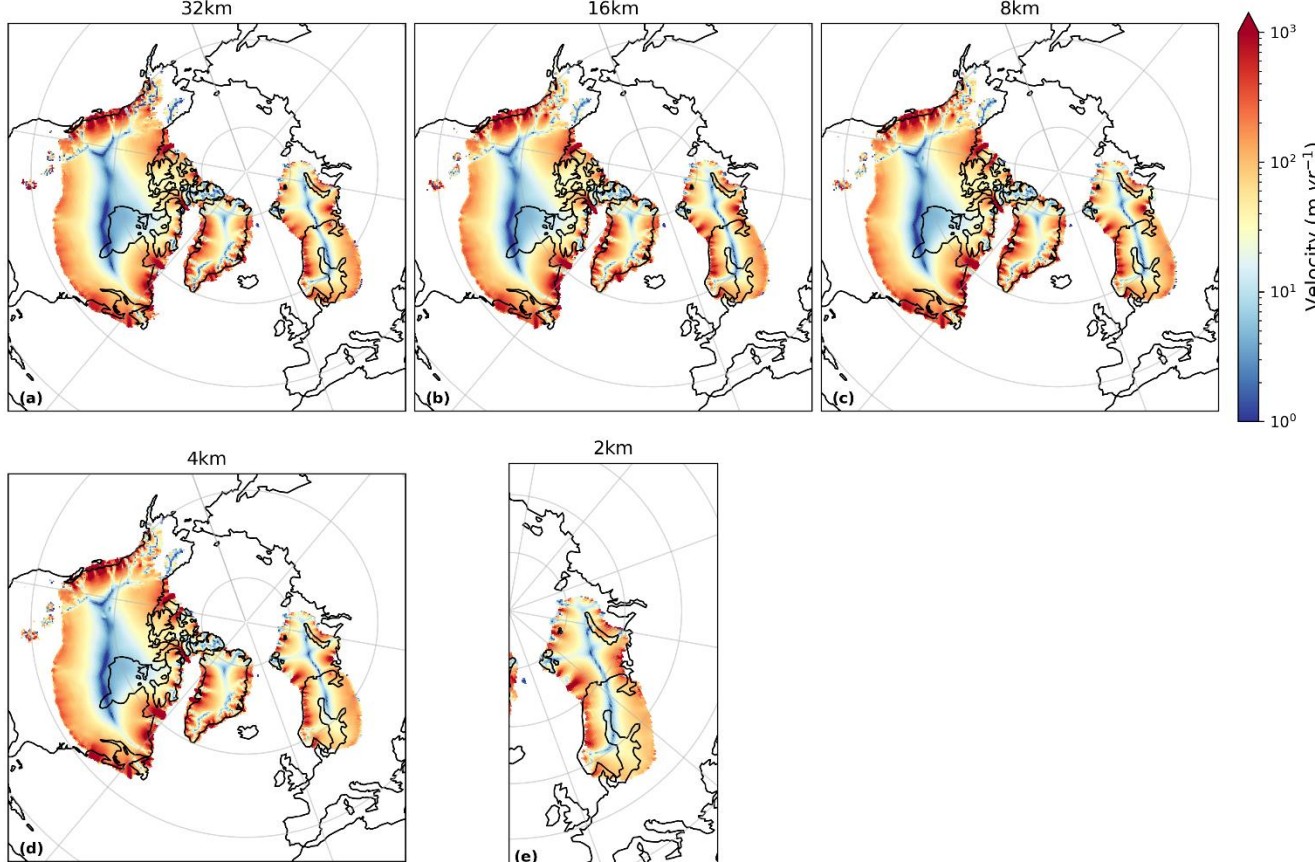

**Figure C2: Ice velocity averaged over the 5000 year simulations using different levels of ice stream refinement. All areas covered by ice were refined to 16 km in panel (b); the ice sheet remains at 16 km and only areas of ice streaming are refined to the finer resolutions indicated in panels (c)-(e). Only the ice streaming across the marine section (BKIS) was refined on panel (e).**



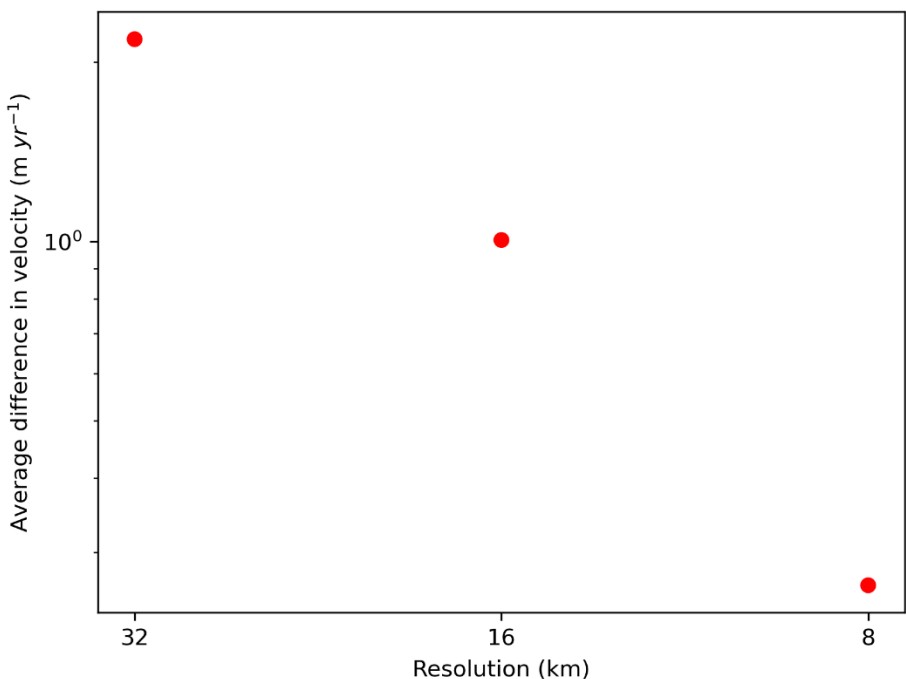

**Figure C3: Difference in ice velocity averaged over the whole ice sheet and 5000 year simulations between the 4km resolution simulation and higher resolutions (8 km, 16 km and 32 km).**

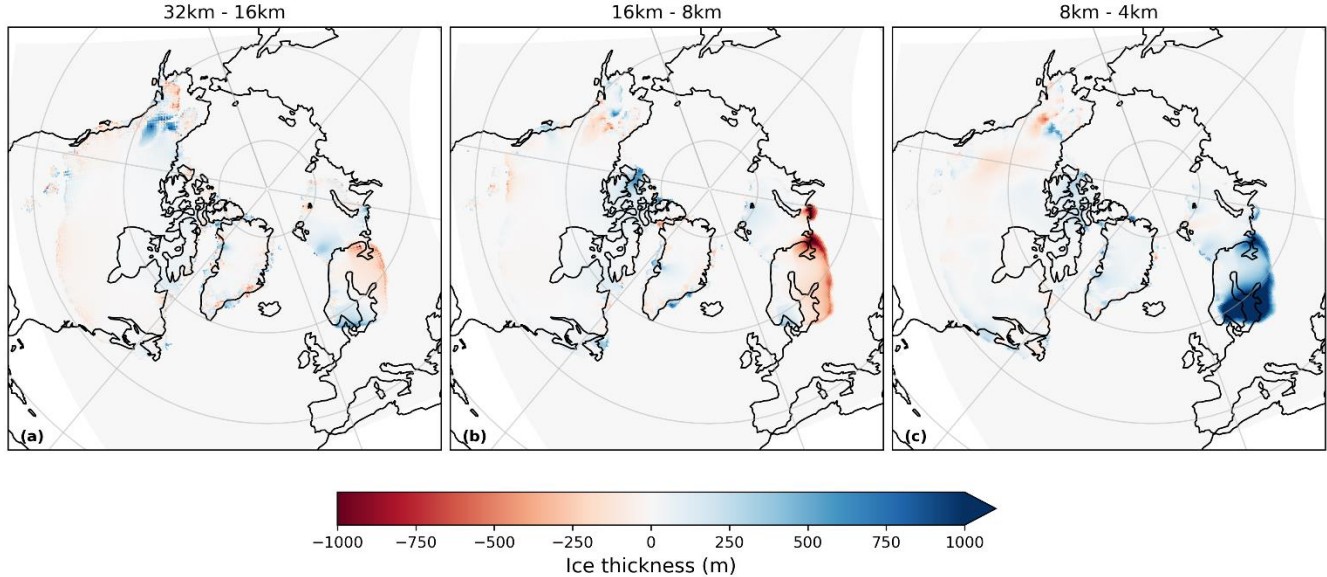

**Figure C4: Difference in final ice sheet thickness between simulations with different levels of refinement**




## Appendix D: Leave-one-out-cross-validation (LOOCV)

Whilst a large proportion of the predicted diagnostics matched the modelled values within the 95 % credible interval, the LOOCV reveals that the Gaussian Process emulator struggled the most with predicting smaller ice sheet volumes and areas. This was especially the case for the PGM Eurasian ice sheet where many of the simulations collapsed due to GIA feedbacks and non-linearities in ice sheet-climate interactions. There is also one obvious outlier in all eight of the diagnostics where the emulator predicted a much higher value than what was actually modelled. This is the same parameter set (xprrk/xpruk) for each.

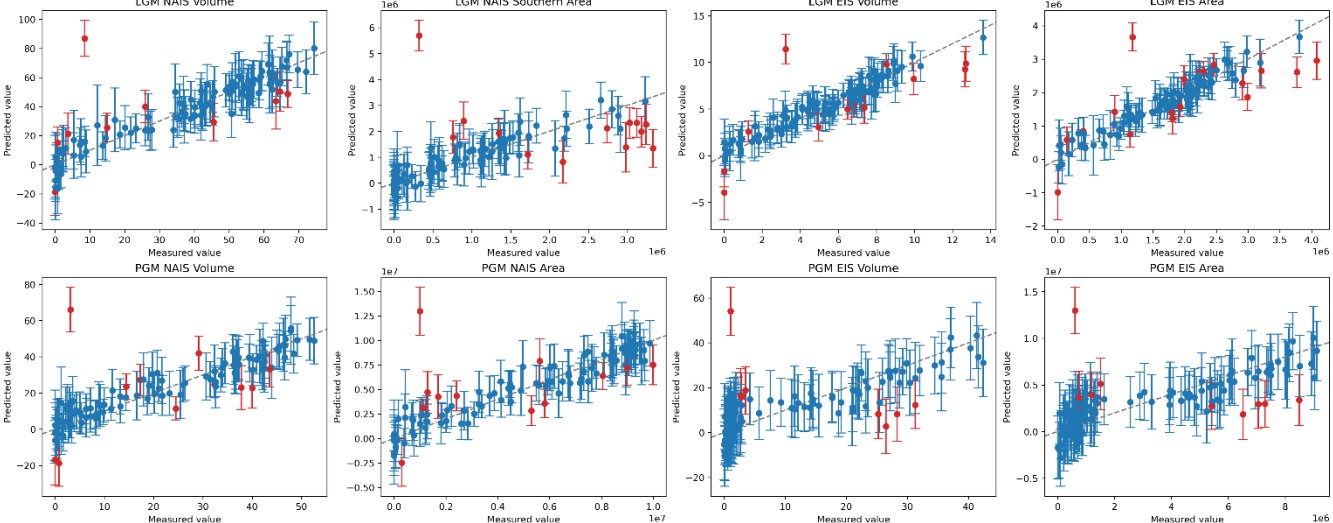

**Figure D1: The results of the Leave-One-Out Cross Validation performed on emulators for the eight diagnostics. The points show the value produced by the numerical model against the value predicted by the emulator for the same sets of input parameters. The line through the centre is the 1:1 line and the error bars show the 95 % credible interval for each point. The points for which the measured value does not fall within the error bars are highlighted in red.**



**Appendix E: Time series of diagnostic variables for NROY simulations**

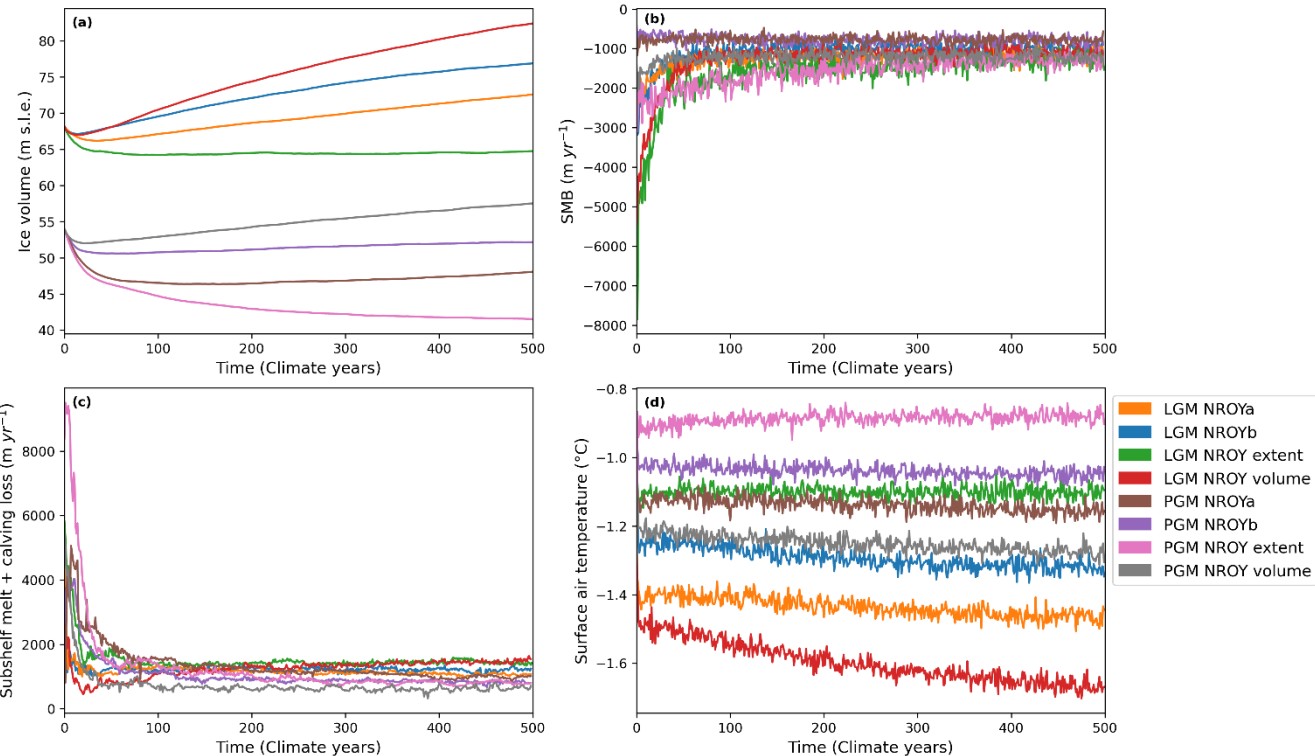


**Figure E1: Time series of variables averaged over North America for the NROY simulations; (a) ice volume; (b) surface mass balance; (c) total sub-shelf melt plus calving mass loss; and (d) surface air temperature.**




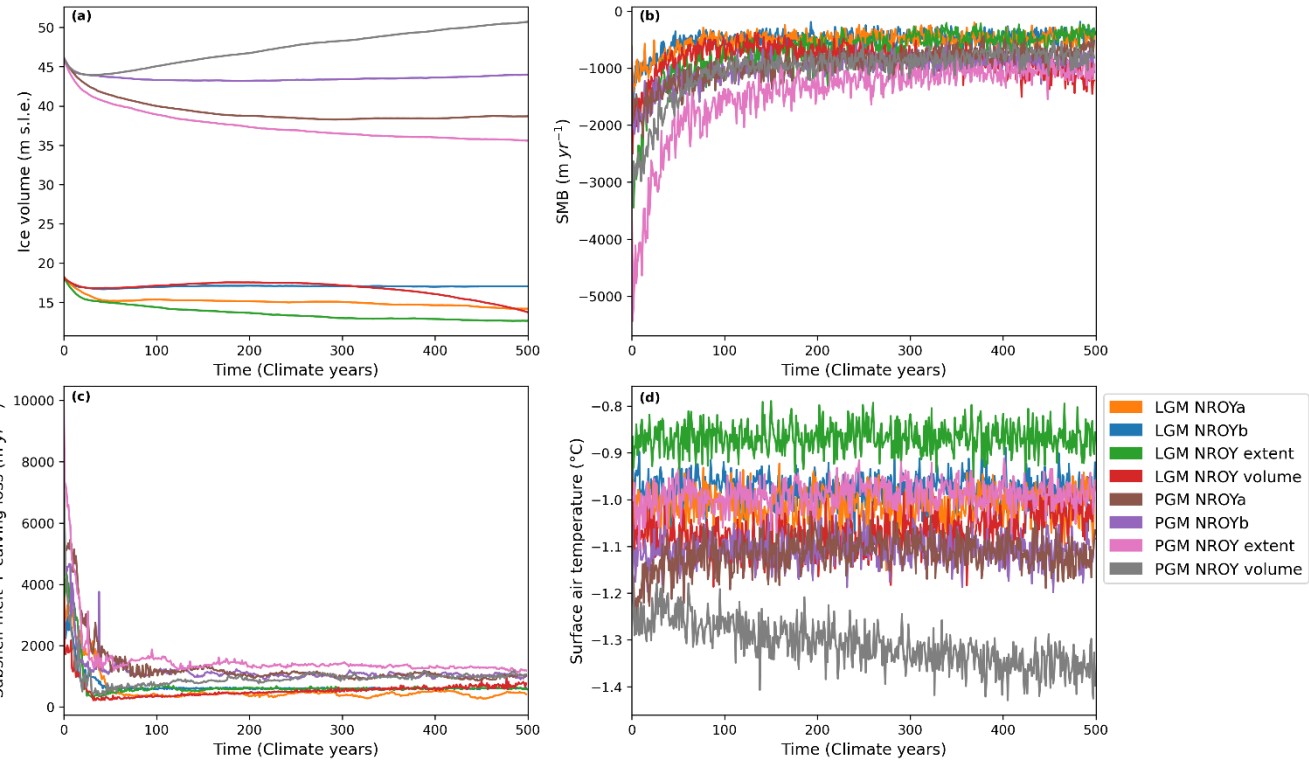


**Figure E2: Time series of variables averaged over Eurasia for the NROY simulations; (a) ice volume; (b) surface mass balance; (c) total sub-shelf melt plus calving mass loss; and (d) surface air temperature.**






**Appendix F: Parameter pairs plot**




**Figure F1: Parameter pair plot of the most influential parameters with the NROYa and NROYb simulations in red, NROY extent**
**simulation in orange, NROY volume simulation in green and the four other simulations that meet the North American ice sheet**
**constraints but not the Eurasian in blue.**





## Appendix G: PGM ice streams

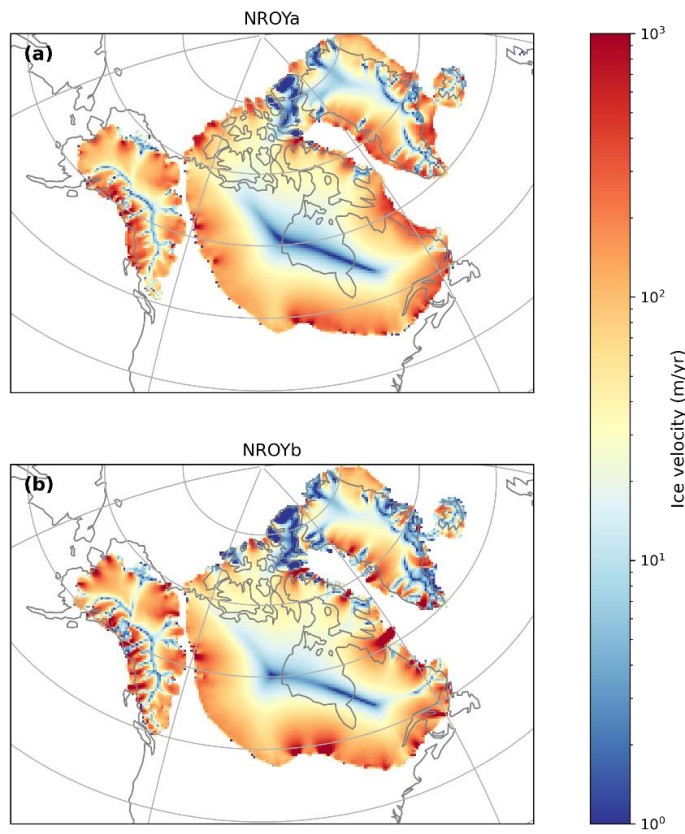

**Figure G1. North American ice sheet ice velocity at the end of the 5000 ice sheet years for the two equivalent PGM NROY simulations**



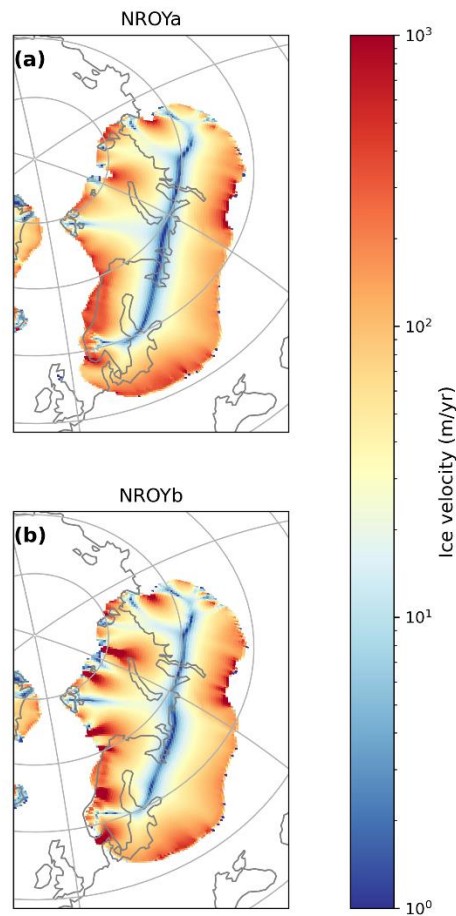

**Figure G2. Eurasian ice sheet ice velocities at the end of the 5000 ice sheet years for the two equivalent PGM NROY simulations**

## Data availability

For this pre-print, the boundary and initial conditions used in this study as well as the full ensemble final year ice sheet model output and volume and extent metrics, climate timeseries for the NROY simulations and final ice sheet model output from the sensitivity tests have been made available to reviewers. All other model output data are available on request.

## Supplement link

## Author contribution

VLP lead the project and performed the majority of the work. VLP, LJG, RFI, and NG designed the simulations, and VLP prepared the initial and boundary conditions, ran the simulations and analysed the results. SC provided technical and scientific



support in the set-up and updating of BISICLES. SST and RSS implemented and tested the elevcon height adjustment parameter. JO provided support on statistical methods including the Sobol analysis and emulation. VLP wrote the manuscript with comments and contributions from all co-authors, with particular contribution from SST on the FAMOUS-ice coupling and elevcon description. LJG, RFI, and NG supervised the project, and LJG acquired the funding.

**Competing interests**

The authors declare that they have no conflict of interest.

**Acknowledgments**

Violet Patterson would like to thank their supervisors and co-authors for their time, support and valuable input on this study. The simulations were run on the high-performance research computing facilities of the University of Leeds, and technical support was provided by Richard Rigby from the Centre for Environmental Modelling and Computation (CEMAC). The authors would also like to thank Oliver Pollard for his help in creating the PGM ice sheet boundary conditions used in this study and his support on the Sobol analysis and GP emulation methodology. Also thank you to Jonathan Gregory for his contribution to developing the elevcon height adjustment parameter.

**Financial support**

This research is primarily supported by the "SMB-Gen" UK Research and Innovation Future Leaders Fellowship (grant no. MR/S016961/1), with Lauren J. Gregoire, Jonathan Owen, and Niall Gandy supported by the award, and Violet L. Patterson's PhD studentship funded by the University of Leeds. Ruza F. Ivanovic and Robin S. Smith's contributions were supported by the RISICMAP19 NERC standard grant NE/T007443/1. Sam Sherriff-Tadano was funded by JSPS Overseas Research Fellowships 202260537.

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
