# Peer review of "Exploring the sensitivity of the Northern Hemisphere ice sheets at the last two glacial maxima to coupled climate-ice sheet model parameters"

_EGUsphere, 2024_

## Author Comment (AC1)

**Response to reviews of 'Patterson et al., Exploring the sensitivity of the Northern Hemisphere ice sheets at the last two glacial maxima to coupled climate-ice sheet model parameters'**

We thank both reviewers for their constructive and helpful comments, which we have used to improve our manuscript. We were happy to hear that the reviewers think our use of coupled climate-ice sheet modelling demonstrates 'valuable improvements' to simulating Northern Hemisphere ice sheets. Following the reviewers' feedback, we propose the following main changes:

1. A re-write of the abstract and introduction to better frame the results of the manuscript, highlight important advances and improve clarity and focus. We will remove the discussion on the reasons for the differences between the LGM and PGM ice sheets as well as reference to investigating climate-ice sheet interactions, as these are not explored in this study. Instead, the introduction will focus on the important influence of ice sheet geometry on glacial/deglacial climate and therefore the need for improved reconstructions of the glacial maximum ice sheets, in particular for the PGM. The explanation of how this study advances on previous studies, through the use of the BISICLES ice sheet model and the inclusion of the Eurasian ice sheet, will also be moved from the Methods to the Introduction.

2. Adjustment of the methods to improve clarity and conciseness. We will move the BISICLES model description into Sect. 2.2, which we will now call 'Ice sheet model description and set-up'. Sect. 2.1 will now contain the climate model and coupling procedure description and be called 'Climate model and coupling'. The BISICLES spin up and sensitivity tests descriptions (Sects. 2.2.1-2.2.3) will be moved to the Appendix.

**Response to Reviewer #1**

"The authors investigate the sensitivity of key parameters in a new coupled climate-ice sheet model, FAMOUS-BISICLES, for simulating the Northern Hemisphere ice sheets at two glacial maxima. They conduct large ensemble simulations and evaluate the results against paleo-evidence of global mean temperature, ice volume and ice extent. To achieve this, they use Gaussian Process emulators and perform a Sobol sensitivity analysis. They finally identify two simulations that satisfy their evaluation constraints for the LGM and PGM."

General comments

"The study demonstrates valuable improvements in using the coupled ice sheet-climate model, FAMOUS-BISICLES, to better simulate the Northern Hemisphere ice sheets during various glacial periods, and highlights key parameters. However, beyond the tuning process and model parameter study, I do not see significant new findings. Therefore, please investigate further the new insights related to simulating the last two glacial maxima. Otherwise, the work may be more suitable for a model development journal."

➢ Our manuscript presents **important scientific advances** with regard to both Quaternary ice sheet evolution and modelling methodology.
  - Prior to this study, there had been very few realistic ice sheet simulations of both the EIS and NAIS at the Penultimate Glacial Maximum and large uncertainties in the shape, size and behaviour of the ice sheets. By simulating the NH ice sheets using coupled modelling, this study provides important data

for interpretation on ice sheet extent and volume, GIA/solid earth, sea level, freshwater fluxes and climate. Our study also provides new and improved PGM and LGM ice sheet configurations that can be used as boundary conditions for climate simulations or initial conditions for coupled climate-ice sheet or sea level simulations of the last two deglaciations.

- Our ensemble history matching approach provides not just one, but four different ice sheet reconstructions for each period and each ice sheet that match reconstructed extent and volume given model and data uncertainty. This provides a means for other studies to evaluate the effect of ice sheet uncertainty on the climate and sea levels.
- Finally, modelling studies that use Latin Hypercube sampling to produce an ensemble of simulations are often criticised for the lack of insight such a method provides in the effects of the parameters varied. By applying a state-of-the-art Sobol sensitivity analysis, we deliver detailed quantification of the effect and relative importance of each parameter. This highlights that the sensitivity to input parameters is not only dependent on the continent/ice sheet, but also varies between glacial maxima.

➢ **The revised manuscript will have edits to the introduction, results/discussion and conclusion to highlight these important advances and improve the framing of the manuscript. This includes the main changes outlined on page 1.**

"The structure and logic of the study need to be improved for greater clarity, and the sentences could be more concise. The Introduction and Methods sections are somehow redundant.

➢ **We will re-write and re-structure the introduction and methods sections as explained on page 1 and will edit the text throughout to improve clarity and conciseness.**

Moreover, I would suggest separating the Results and Discussion section to improve clarity and organization."

➢ Separating the results and discussion would lead to a disconnect between the results and the explanation for the changes observed. **We have thus decided to keep the current structure for greatest clarity.**

"In the introduction, the authors discuss various uncertainties in reconstructing the PGM ice sheets and the possible reasons for the differences between the PGM and LGM ice sheets. However, it is unclear how these discussions are linked to the final conclusion, as the reason for the differences between the PGM and LGM ice sheets are not investigated in this study."

➢ The reasons for the differences between the PGM and LGM were investigated in Patterson et al. (2024). **To improve the framing of this manuscript we have removed the paragraph on this topic from the manuscript and rewritten the introduction to improve the framing of this work, as detailed on page 1.**

"I'm not sure if it is appropriate to use "coupled climate-ice sheet" in the title, given that the ocean is prescribed, and the active components are only the atmosphere and the ice sheet."

➢ **We will change 'coupled climate-ice sheet' to 'coupled atmosphere-ice sheet' and give the manuscript a new title** along the lines of 'The Eurasian and North American

ice sheets at the Last and Penultimate glacial maxima: coupled atmosphere-ice sheet model sensitivity and calibration' to better capture the contents of the manuscript.

Specific comments

"Line 16-18: The background introduction in the abstract is somewhat redundant and somewhat off-topic. The authors suggest that the answer likely lies in "the different orbital configurations between the two periods" and "climate-ice sheet interactions". However, the manuscript does not address the differences in orbital configurations or climate-ice sheet interactions."

> **We agree and this section will be removed from the abstract and introduction as it is off topic and these sections will be re-written as explained on page 1.**

"Line 20: "better understand how NH ice sheets interact with the climate". However, the manuscript does not examine the interaction between the ice sheets and the climate. Instead, the parameters investigated in the model mainly focus on downscaling or factors that influence ice sheet surface/basal processes."

> **See response to previous comment**

"Line 26-30: What does the statement "… we find two simulations …" indicate? Does it suggest that the parameter criteria for simulating the LGM/PGM ice sheets are too strict? Otherwise remove this sentence.'

> The judgement of whether criteria are too strict or not should be with respect to what the chosen simulations are used for, not how many simulations are ruled out in an ensemble. This is because the way parameters are sampled is based on subjective choice and controls the proportions of ruled out simulations. Our previous work with FAMOUS-ice had already revealed that finding combinations of parameter values that produce realistic ice extent during glacial times is challenging, due to strong climate-ice sheet interactions. However, finding two parameter combinations that produce plausible results for both time periods and both ice sheets is a very good outcome. More simulations could be found by running further waves of experiments. This is computationally expensive and time consuming and was not required for our purposes.
> **We will edit the results section to include this discussion and the revised abstract will be improved.**

"Additionally, please clarify the following sentence by specifying which parameters are more sensitive to which ice sheet or time period."

> **We will clarify this in the abstract the best we can within the word limit.**

"Line 180 and beyond: It is somewhat odd that you claim BISICLES is a model well-suited to simulate the past evolution of marine ice sheets, yet in the experiment setup, the ocean is prescribed and not investigated."

> Yes, we prescribe the ocean forcing to the atmosphere model and the sub-shelf melt forcing to the ice sheet, but we investigate the Eurasian ice sheet, which is a marine ice sheet in the Barents Sea sector. We also investigate the dependence of the Eurasian ice sheet on parameters controlling sub-shelf melt and grounding line migration. We stand by our comment that BISICLES, as an ice sheet model, is very

well suited to simulate Marine ice sheets (ice sheets that are based mostly below sea level and have extensive grounding lines). Furthermore, our methods are appropriate for simulating marine ice sheet evolution. Most simulations of Marine ice sheets, whether modern (eg. ISMIP) or paleo, run with BISICLES, PISM, ISSM or other ice sheet models do not include an interactive ocean, but are ice-sheet only simulations driven by prescribed sub-shelf melt as we do here. The feedback between the ocean and the submerged parts of the ice sheet are difficult and expensive to model interactively and we do not believe that including this component would improve our study.

➤ **We will work on justifying this point in the methods.**

"Figure 1a-b: Please also include the difference in SST between the LGM and PGM."

➤ **We will update Figure 1 to include this extra panel**

"Line 413-416: Are the differences in the mean values for the different ice sheets at the LGM or PGM likely due to the different initial conditions? The same question applies to the spatial pattern (Line 418-421). Additionally, please elaborate on what these calculations indicate."

➤ Yes, the relative difference in volume and extent of the LGM and PGM ice sheets is likely largely due to the different initial conditions as shown in Patterson et al., (2024). These calculations are included to give an overview of the equilibrium ice sheets produced by the ensembles. The results are a consequence of the initial ice sheets, climate forcing and parameter values used.

➤ **We will clarify this in the text.**

"Figure 6-7: Maybe use a different colormap to display the ice thickness pattern. Currently, it is difficult to identify the simulated margin."

➤ **We will add a blue contour line to indicate the extent of our simulated ice sheets**

"Line 517-518 and beyond: Please explicitly indicate what "tgrad" and "drain" refer to, and explain what their indications are. The same applies to the rest of the manuscript (e.g., Line 558, 544, 562…)"

➤ **We will add what these parameter names refer to in brackets after their first mention in the results**. Readers can refer to the Table 1 for a more in depth explanation.

**Response to Reviewer #2**

"The authors examined the sensitivity of several parameters in the FAMOS-BISICLES Atmosphere-Ice Sheet model, applying their system to produce simulations of the NHIS at last two glacial maxima. While I find that the study provides an interesting parameter study and an extensive evaluation of the results against paleo-evidence, there are several comments which should be addressed before this paper can be considered for publication."

General Comments

"I find the language of the manuscript to be rather verbose. A simplification would be beneficial"

➤ **The manuscript will be edited to shorten and simplify the text.**

"While comparing simulated ice sheets to previous studies, the authors lack discussion of their broader applicability beyond their modeling framework. It's important to consider if ice sheet volume and extent sensitivity to albedo parameters is consistent across models. While it would be beyond the scope to turn this into a multi-model comparison, explicitly discussing limitations and potential translations to other approaches would strengthen the manuscript."

> ➢ Sensitivity to model parameters is model dependent, but we show that sensitivity not only depends on the continent/ice sheet but also the glacial maximum, even though the time periods have similar climates. Our results also suggest a relation between the sensitivity to some of the parameters and the size of the ice sheets. This highlights the mechanisms that are important for different periods/ice sheets and these results are likely to hold for other models.
> ➢ **We will edit the results section to include this discussion.**

"Importantly, the use of the term 'coupled climate-ice sheet' in both the title and the text is misleading and fundamentally incorrect. The model lacks a fully interactive ocean component, with only the atmosphere and ice sheet being active. This omission means it does not qualify as a true climate model, and the terminology is inappropriate. Referring to it as such overstates the model's capabilities and misrepresents its actual functionality. The authors do, however, mention this limitation and that this shall be the subject of future work."

> ➢ **We will change 'coupled climate-ice sheet' to 'coupled atmosphere-ice sheet' in the title and text.**

"The Methodology section needs some reorganization. Several auxiliary experiments that describe the sensitivity of ice dynamics are presented, but these are ice-sheet-only experiments and have only a tangential impact on the coupled system, where different parameters are evaluated. While these additional experiments are undoubtedly necessary and prudent to properly contextualize the ice sheet simulations within the coupled ice/climate system, I believe they could be moved to an appendix. This would allow the reader to focus on the main findings related to the coupled system."

> ➢ **Thank you for the suggestion, we will revise the manuscript in line with this advice as detailed on page 1.**

Specific Comments

"Introduction Section: I find that this could use more focus. The introduction and background suggest that orbital configuration and climate/ice interactions are responsible for the differences seen in the PGM and LGM ice sheet geometries. However, neither orbital differences nor climate-feedbacks are explicitly discussed in the remainder of the manuscript. I would here rather focus on what you are parameterizing."

> ➢ **We will re-write the introduction, as described on page 1, to improve the focus.**

"Methods: While it's evident that this isn't the primary focus of your study, I find it unfortunate that the potential influence of the Antarctic ice sheet on the climate during the two glacial maxima isn't discussed at all."

> ➢ Our manuscript focuses only on the Northern Hemisphere ice sheets. The Antarctic ice sheet is outside the scope of this study.

"Lines 145-160: It is unclear from the description how surface fluxes of ice mass (meltwater) are handed back to the climate model. It seems that the mass fluxes calculated by the atmosphere are interpolated onto the ice sheet grid and used for SMB, however, the coupling step on the way back to the climate only considers ice extent and orography. Is this correct?"

> ➤ This is correct; see the response to the next point.

"Line 204: What about basal melting discharged over land at the edge of the ice sheet? This is simply removed? This would imply that the model has a mass leak."

> ➤ Routing of surface and basal meltwater matter for the hydrological cycle and ocean circulation. Since our simulations do not include an interactive ocean, there is no need to close the climate system hydrological cycle (ie. Atmosphere-ocean-cryosphere-land).
> ➤ **We will add this to the text as a reminder to the readers who are more accustomed to coupled atmosphere-ocean-ice sheet modelling.**

"Lines 562-621: I would be curious in this discussion to see some mention of implications of the (fairly coarse) atmospheric model resolution. The authors mention that precipitation is not downscaled in their approach, which may be a reasonable assumption to make. It might be beneficial to see how this effect plays out with an atmosphere model with finer spatial resolution (or alternatively, references to literature describing these implications)"

> ➤ **We will add comments on the role of horizontal atmospheric resolution and precipitation downscaling on the simulated ice sheet geometries in the discussion.**

"Conclusion Section: The authors switch tenses. For example: "We ran ensembles…" coupled later with "Through Gaussian Process emulation… we find…". I believe this should instead stay consistent throughout the text."

> ➤ **We will fix this**

"Figure 1: An anomaly plot vs. PI for SST might be clearer, as well as a relative difference between the two glacial states"

> ➤ **We will update Figure 1 to show the LGM and PGM SSTs as an anomaly from PI and add an additional panel for the difference**

"Figure 2: This could be combined into a single map for the entire Northern hemisphere. Presenting two separate panels gives the false impression that you have two distinct, separate ice sheet simulations running (one for each domain). I also think it would be more consistent to plot the coastlines as seen in the model here, rather than modern ones."

> ➤ Figure 2 discussed here shows the reconstructions of ice sheet extent. A single figure of the full northern hemisphere ice sheet domain is shown in Figure 1, and Figures 4-7 show results of the two ice sheets on one plot. The manuscript's text clearly describes the ice sheet domain. We therefore disagree with the reviewer's point that "Presenting two separate panels [in Figure 2] gives the false impression that you have two distinct, separate ice sheet simulations running (one for each domain)." In fact, we argue that plotting the ice sheet reconstructions for North America and Eurasia on two separate panels makes it clear that they are from different sources.
> ➤ Plotting the coastlines used by the climate model would not be particularly useful on this figure since the ice sheet reconstructions shown are independent from our

modelling. They do not provide reconstructions of the coastlines, so it is clearer to plot them with modern-day coastlines.

"Figure 4: A colormap with discrete level boundaries would be nicer here, in my opinion."

➢ **We will update this figure to make the colour scale easier to read**

"Figure 5: The white-gray colormap is difficult to read and understand, I would suggest a different colormap here."

➢ **We will update this figure to change the colormap to improve clarity**